# Systematic review of generative adversarial networks (GANs) in cell microscopy: Trends, practices, and impact on image augmentation

**Duway Nicolas Lesmes-Leon**[1,2]*, **Andreas Dengel**[1,2], **Sheraz Ahmed**[2]

**1** University of Kaiserslautern-Landau (RPTU), Kaiserslautern, Germany, **2** DFKI: German Research Center for Artificial Intelligence GmbH, Kaiserslautern, Germany

* dlesmesleon@gmail.com; lleon@rhrk.uni-kl.de

**Data availability statement:** All relevant data are within the manuscript and its Supporting Information files.

## Abstract

Cell microscopy is the main tool that allows researchers to study microorganisms and plays a key role in observing and understanding the morphology, interactions, and development of microorganisms. However, there exist limitations in both the techniques and the samples that impair the amount of available data to study. Generative adversarial networks (GANs) are a deep learning alternative to alleviate the data availability limitation by generating nonexistent samples that resemble the probability distribution of the real data. The aim of this systematic review is to find trends, common practices, and popular datasets and analyze the impact of GANs in image augmentation of cell microscopy images. We used ScienceDirect, IEEE Xplore, PubMed, bioRxiv, and arXiv to select English research articles that employed GANs to generate any kind of cell microscopy images independently of the main objective of the study. We conducted the data collection using 15 selected features from each study, which allowed us to analyze the results from different perspectives using tables and histograms. 46 studies met the legibility criteria, where 23 had image augmentation as the main task. Moreover, we retrieved 29 publicly available datasets. The results showed a lack of consensus with performance metrics, baselines, and datasets. Additionally, we evidenced the relevance of popular architectures such as StyleGAN and losses, including Vanilla and Wasserstein adversarial losses. This systematic review presents the most popular configurations to perform image augmentation. It also highlights the importance of design good practices and gold standards to guarantee comparability and reproducibility. This review implemented the ROBIS tool to assess the risk of bias, and it was not registered in PROSPERO.

## Introduction

The study of microorganisms is essential for progress in medicine, as it leverages the development of drugs, treatments, and diagnosis of diseases, allowing mankind to improve not

**Funding:** This study was partially funded by Sartorius AI Lab (SAIL), a collaboration between Sartorius (https://www.sartorius.com) and the German center for artificial intelligence (DFKI) (https://www.dfki.de). The funders had no role in study design, data collection and analysis, decision to publish, or preparation of the manuscript.

**Competing interests:** The authors have declared that no competing interests exist.

only their quality of life, but also their relationship with the environment [1]. Microorganisms cannot be seen with the naked eye, which had been a major limitation in the past. Fortunately, thanks to cell microscopy, researchers now count with several available tools that have relieved this limitation considerably.

Cell microscopy imaging is the set of techniques used to visualize microorganisms, which allows researchers to study anatomy, dynamics, and interactions between microorganisms. There are different microscopy techniques that have been developed to highlight specific features. Some of the most popular techniques are optical, fluorescence and electron microscopy. However, each technique has intrinsic limitations; some have limited magnification, some require killing and fixing the samples, and others require very specialized tools, making them inaccessible. Moreover, The preparation of the data carries its own challenges, including expensive and long sample preparation, environmental conditions that are not always reproducible, a limited time window to visualize important events, ethical considerations depending on the source of the data, and time-consuming and prone-error analysis carried out by experts. The combination of all these factors contributes to a significant impairment in the production of both raw and annotated data.

In the last years, deep learning (DL) has provided the field of computer vision with several alternatives to alleviate the challenges mentioned earlier. Training DL models in different tasks has several benefits that include decreasing image evaluation times, capturing information that is not easily seen to the naked eye, releasing specialized personnel from mechanical tasks, and simplifying their complexity. It is well known that model performance is proportional to the amount of data available to train; normally a significant amount of training data is required to achieve the best model generalization possible, and even though data is not a limitation for most fields, specialized domains such as cell microscopy struggle in this regard [2].

Moreover, the microscopy imaging domain has features that differentiate it from the rest of specialized domains. The main characteristic of microscopy is that the number of objects present in an image can vary greatly, reaching even hundreds of objects per image, introducing important considerations when training DL models. The number of pixels representing a single object can be minimal in microscopy imaging compared to natural images (objects tend to take up most of the image), reducing the model capability to capture local features of the objects of interest. Isolating the cells might not be beneficial in all cases since the resolution would be too low and the cell distribution over the image could bring meaningful information about the cell-to-cell interactions as well. There would be a high amount of cell samples but in a limited number of images due to the intrinsic limitations of microscopy. Additionally, densely populated images lead to recurrent object overlapping, which could also impair the model performance. In the case of anomaly samples, very few cells might present the anomaly within the image, leading to an underrepresented class at both pixel and cell level.

Besides the general factors of the cell microscopy domain, an important consideration to always keep in mind is that reproducing style features is a difficult task. Each image has unique characteristics that will always differ from the rest even if the images come from the same biological sample, e.g., the color, illumination, position, sampling protocol, and the sampling device. Such differences are, typically, not a constraint to the analysis of experts; nevertheless, DL model generalization can be highly affected by them [3]. Normalization and style transfer techniques are essential in the preprocessing steps to ensure robust models that focus on the most essential features of the images, even more when there is more than one data source.

Fortunately, generative modeling is an area of machine learning whose task is to understand the distribution of data to produce synthetic samples resembling the real data distribution. Generative adversarial networks (GANs), a well-established generative model and the focus of this study, comprise a family of DL algorithms based on the intuition of training two neural networks (NNs) simultaneously in a competitive fashion [4]. In computer vision, the evolution of GANs allowed researchers to propose promising solutions to several tasks comprising image classification, segmentation, augmentation, enhancements, and domain translation [5]. There are reviews analyzing GANs, their general foundation, and applications [5,6], and specialized reviews of GANs in medical imaging [7–11]. Although some of these reviews compile microscopy image datasets, to the best of our knowledge, there is no published review focused on cell microscopy generation.

We present a systematic review of GANs using cell microscopy imaging for image augmentation. The objectives of this work are:

- Analyze the studies using GANs to perform cell microscopy image augmentation.
- Analyze popular public available datasets to train generative models for cell microscopy image augmentation.
- Identify the most popular architectures, losses, and methods when using GANs in the field.
- Identify common practices of experimental design related to image augmentation.

We first screened different databases and selected relevant publications based on legibility criteria to analyze and summarize them to discover the main trends, challenges, and limitations present in the field. Moreover, we analyzed the publicly available datasets used in the selected publications and summarized their principal characteristics. Finally, we briefly discuss the most representative GAN architectures and definitions for those readers with a biological background or enthusiastic researchers that are introducing themselves to the field of GANs. The main contributions of this work are:

- A comprehensive compilation of 46 chosen research studies sourced from 5 distinct databases, encompassing several features aimed at facilitating a thorough comparative analysis.
- A compilation providing detailed descriptions of the publicly accessible datasets utilized within the studies considered in this review.
- Important notions and considerations in the experimental design of generative modeling studies that are essential to produce research with robust methodology and results.
- A concise list of good practices when implementing GANs for image augmentation in cell microscopy.

## Materials and methods

This review was carried out following the guidelines of the Preferred Reporting Items for Systematic Reviews and Meta-Analyses (PRISMA 2020) [12] as close as possible (S3 Table). Even though PRISMA 2020 is mainly designed to structure systematic reviews related to medicine, the results of this review could lead to ideas in the context of DL that could benefit the health field. Additionally, we believe that implementing these guidelines enhances the process and outcomes of a paper review, independent of its context. This systematic review is not registered in PROSPERO [13], and it neither has nor follows any review protocol.

**Eligibility criteria and search strategy.**  The only source of information selected for the review are papers written in English whose main topic was the implementation or

design of GANs. Additionally, the purpose of the GAN must be image augmentation of cell microscopy. Even though I2I translation could be considered an image augmentation method, we defined image augmentation as transforming random noise into a generated sample exclusively (as in the traditional GAN definition). This definition rejects approaches such as domain translation, enhancement, or normalization, which typically rely on paired or domain-specific data. Such data may not always be available in microscopy, an image domain where data acquisition is often limited due to experimental, biological, or financial constraints. In contrast, noise-based GANs offer a general solution for synthetic sample generation in low-data regimes, making them especially relevant in this context. Moreover, limiting the review to these architectures allowed for a more focused and structured analysis, avoiding dilution across multiple GAN subtypes while still addressing a foundational problem in microscopy: generating diverse samples from scarce data. Papers are legible as long as they perform image augmentation independently of whether it was not the study goal, i.e., an image classification study that used a GAN architecture to augment the dataset is legible and therefore selected for this review.

The selected databases are IEEE Xplore, ScienceDirect, PubMed, arXiv, and bioRxiv. We retrieved studies without limit in the year of publication, meaning that we will consider any publication from the introduction of GANs in 2014. Additionally, we did not include publications cited in the retrieved papers or publications from different sources, considering that our search criteria include the year of publications of GAN. Table 1 summarizes the query search and other search parameters used in each database.

A single person was responsible for the whole paper selection process. After removing duplicates, we examined the title and abstract to know if the publication met the eligibility criteria; in case of doubts, the methods and results sections were checked. Papers were considered equally independent of their performance assessment (quantitative, qualitative, specific metric) as long as the publication fulfills the eligibility criteria. In case of missing data (due to inaccessible manuscript), the study was not considered for the analysis. Finally, we did not implement any risk of bias tool in the selected studies due to the nature of the field. However, we implemented the ROBIS tool [14] to assess the risk of bias of this systematic review.

**Synthesis methods.** Considering that we aim to verify the main applications of GANs in cellular imaging for image augmentation, we decided to use a table as the dominant strategy to summarize the data. The table features are publication, GAN loss, generator (architecture), discriminator (architecture), Performance Metrics, scores, model baselines, data type,

**Table 1. Advanced search query parameters used as search strategy.**

| Database | Field | Query search | Areas/Servers | Last visit |
|---|---|---|---|---|
| BioRxiv | Title OR abstract | "Generative adversarial network" | bioRxiv, medRxiv, bioRxiv and medRxiv | 08/04/2025 |
| aRxiv | Abstract | ("Generative adversarial network") AND (cell OR microscopy OR histology) AND (augmentation OR generation OR synthesis) | "Computer Science (cs)", "Mathematics (math)", "Quantitative Biology (q-bio)", "Statistics (stat)" | |
| ScienceDirect | Title, abstract or author-specified keywords | | Research articles | |
| PubMed | Title/Abstract | | N/A | |
| IEEE Xplore | Abstract | | | |

dataset, training type, ablation study, code (availability), designed/modified (D/M), image augmentation as the main task, and notes. Additionally, we designed bar plots when possible to facilitate data visualization.

Depending on the table feature, we considered different conventions to facilitate the data analysis. In the case of data type, we use optical microscopy to represent all datasets acquired with all kinds of light microscopy techniques, including histology or wide-field microscopy. Similarly, the fluorescent microscopy tag includes immunofluorescence images. GAN loss, generator, and discriminator features are simplified using the most representative definitions to minimize the number of possible classes. We decided to keep the performance metrics and scores original, since this study aims to identify the most popular metrics employed in cell imaging. Code, D/M, and image augmentation as main task features are binary classes. The first one indicates if the publication has a URL linking to the original implementation; the second one tells if authors created, modified, or redesigned the model architecture or losses from older publications; and the last one tells if the publication goal was to perform image augmentations or if it was an intermediate process. To further synthesize the results, we grouped the studies based on their adversarial loss and briefly discussed their applications, implementations, and common things.

We summarized the publicly available datasets used in the studies that met the legibility criteria similarly to the table described before. The features extracted from each dataset are microscopy modality, number of samples, image resolution, the task for which the dataset was designed for, other published applications of the dataset, a binary feature to tell whether the dataset is annotated, cell type, URL, and notes. Moreover, we included a detailed description of each dataset.

## Results

Our methodology retrieved a total of 492 publications, and 46 met the eligibility criteria excluding duplicates: 14 (30.43%) from ScienceDirect, 11 (23.91%) from IEEE Xplore, 11 (23.91%) from arXiv, 6 (13.04%) from PubMed, and 4 (8.70%) from bioRxiv. Three publications from PubMed had to be excluded since they were inaccessible. Detailed information on the selection process is described in Fig 1.

We decided to include preprint archives (arXiv and bioRxiv), given that eleven out of the 15 retrieved preprints have been published in peer-reviewed journals or conference proceedings already, leaving only four studies as preprints [15–18]. While a more thorough assessment of preprints is generally advisable, we concluded that our review protocol captures the most relevant and meaningful contributions. Notably, the only preprint featuring both a code release and architectural modifications was published in 2025 [18]. This study investigates alternative discriminator designs and loss functions for federated learning in hospital settings, with the aim of preserving data privacy while enhancing classification performance through data augmentation. The authors report that commonly used generative model benchmarks do not always correlate with perceived image quality or the performance of downstream tasks.

For the sake of simplicity, we define as Vanilla GAN the first implementation made by Goodfellow et al. [4]. Moreover, we divide the GAN into two parts: its loss and its architecture. In this way, a GAN can be implemented with Vanilla adversarial loss and convolutional neural network (CNN) architectures, such as Deep Convolutional GAN (DCGAN) [19].

Considering that we extracted several features from the studies, we decided to analyze the studies from different perspectives, using individual features as points of views. Fig 2 visually summarizes the distribution of the studies based on the features we will focus on.

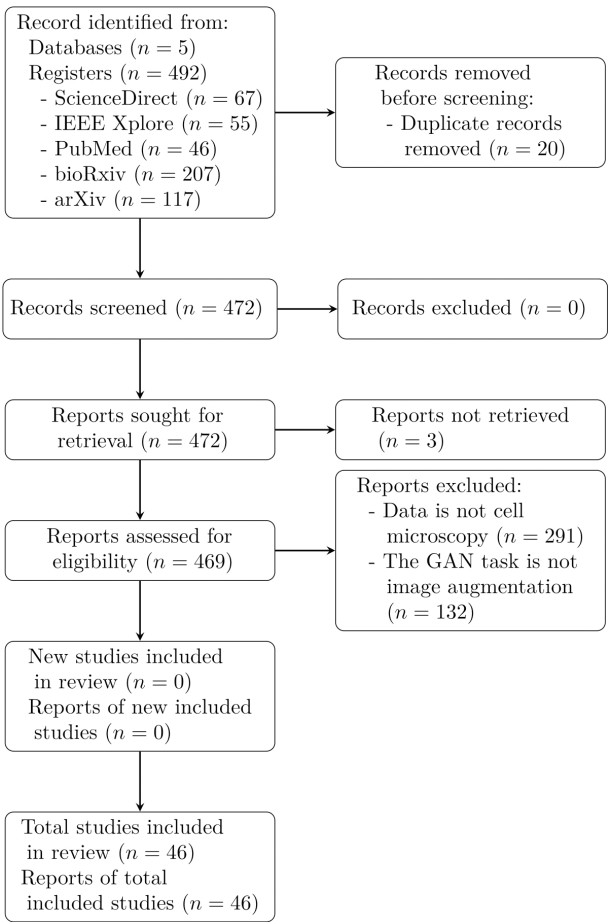

**Fig 1. PRISMA search strategy flow diagram.**

## Year of publication

We observed in Fig 2A an increased interest in data augmentation from 2020; at least six GAN implementations per year since 2020. In 2025, the present year, there are two studies in the first trimester, suggesting that even with more model alternatives, GANs still have a role in data augmentation. The increase in 2018 could be attributed to the gained interest in the field due to the introduction of Pix2Pix [20] and CycleGAN [21] in 2017. The oldest publication on cell image augmentation is from 2017. Considering that a single publication used the Vanilla GAN architecture [15] and that DCGAN was published in 2016, we suppose that Vanilla GAN (multilayer perceptron-based architecture) was not powerful enough to produce realistic microscopy images before 2016 when DCGAN was published.

## Microscopy modalities

Fig 2B displays the microscopy modalities of the employed datasets. From there, it is possible to see the dominance of optical and fluorescence microscopy, present in 30 and twelve studies, respectively. The remaining three modalities sum up the four studies left, suggesting possible data acquisition limitations for different reasons. Although there is a clear interest in the two microscopy modalities mentioned before, we would like to highlight that the dataset feature

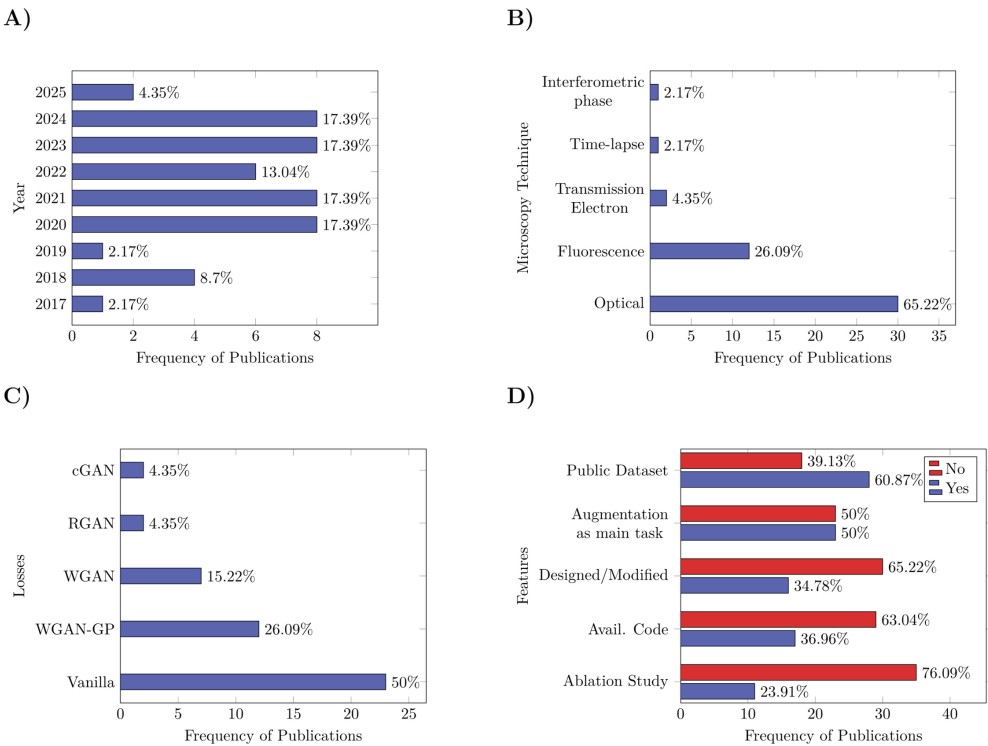

**Fig 2. Histograms from the selected studies based on the features used as point of view. (A)** Yearly publication distribution of GANs used for cell microscopy image augmentation. **(B)** Distribution of microscopy modalities from the datasets used in the selected publications. **(C)** Distribution of adversarial losses. **(D)** Distribution of reproducibility and other features.

is one of the most variable among the classes; the majority of the selected studies use a unique dataset, impeding the comparison between them.

## Adversarial losses

The goal of GANs is to learn the probability distribution of a training dataset and use such distribution to generate new synthetic data that cannot be differentiated from the original dataset [22]. A GAN is traditionally composed of a generator that tries to learn the training data distribution while the discriminator has to differentiate between synthetic and authentic data [4]. The training procedure of GANs can be conceived as a competition where the generator tries to fool the discriminator with realistic samples while the discriminator is trained to detect the generated ones.

Based on the first GAN publication [4], GANs has three domains: noise ($Z$), real data ($X$), and synthetic data ($X'$), and its formal definition is described as follows: with input noise $z \in Z$ following the probability distribution $p_z$ and training data $x \in X$ with probability distribution $p_{data}$, two differentiable functions (generator and discriminator) represented as artificial neural networks (ANNs) are trained simultaneously.

The generator $G$ takes $z$ as input and transforms it into a synthetic sample $x' \in X'$ that follows the generated probability distribution $p_g$. The discriminator $D$ takes input samples from both $X$ and $X'$ and returns a scalar representing the probability that its input belongs to $X$. If,

e.g., $D = 0$, then the discriminator concludes that such input has a probability of 0% to belong to $X$, making it a synthetic sample.

The goal of $G$ is to approximate $p_g$ to $p_{data}$ as much as possible using the information that $D$ retrieves. Thus, it will be harder for $D$ to find the synthetic samples once $G$ is well-trained. An ideal case for $G$ is that $D = 0.5$ independently of its input. The GAN architecture is illustrated in Fig 3.

GANs have had a high development since first introduced by Goodfellow et al. in 2014 [4]. Several loss variants, architectures, and even additional components have been proposed to boost the performance of GANs. Table 2 summarizes the adversarial losses retrieved in this review, including their loss function, what it approximates to (dynamics), and key attributes to facilitate the comparison of them in experimental designs. For those readers interested in their formal definitions, demonstrations, training dynamics, and theoretical implications, we

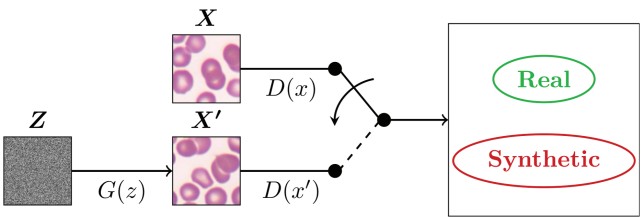

**Fig 3. General GAN architecture.** Domains are represented as squares and model components as arrows. Generator $G$ transforms random noise $z$ into synthetic sample $x' \in X' \sim p_g$ and discriminator $D$ classifies whether its input belongs to $p_{data}$ or $p_g$. Figure design based on Goodfellow et al. [4].

**Table 2. Summary of adversarial losses.**

| Loss | Equation | Approximates to | Attributes |
|---|---|---|---|
| Vanilla | $\min\limits_{G} \max\limits_{D} \mathcal{L}(G,D) = \mathbb{E}_{x\sim p_{data}}[\log D(x)]$ $+ \mathbb{E}_{z\sim p_z}[\log(1 - D(G(z)))]$ | Jensen Shannon Divergence ($\mathcal{D}_{JS}$) with optimal discriminator | • Robust against instability due to gradient exploit. • Instability due to gradient vanishing in case of disjoint supports. |
| Least-Square | $\mathcal{L}_{LS}(D) = \frac{1}{2}\mathbb{E}_{x\sim p_{data}}[(D(x) - b)^2]$ $+ \frac{1}{2}\mathbb{E}_{z\sim p_z}[(D(G(z)) - a)^2]$ $\mathcal{L}_{LS}(G) = \frac{1}{2}\mathbb{E}_{z\sim p_z}[(D(G(z)) - c)^2]$ | Pearson $\chi^2$ divergence ($\mathcal{D}_{\chi^2}$) with optimal discriminator | • The least-square objective encourages $G$ to push the generated samples to the decision boundary of $D$. • Instability due to gradient vanishing in case of disjoint supports. |
| Wasserstein | $W(p_1, p_2) = \sup\limits_{\|f\|_L \leq 1} \mathbb{E}_{x\sim p_1}[f(x)] - \mathbb{E}_{x\sim p_2}[f(x)]$ | Wasserstein-1 distance with K-Lipschitz | • This objective measures the distance between distributions, making it robust against disjoint supports. • The discriminator is renamed as critic $C$ since it will no longer perform classification, $C$ outputs now a distance. • The critic output is correlated with image quality, a property that allows to monitor the training progress. |
| Relativistic | $\mathcal{L}_{rel}^D(C) = \mathbb{E}_{(x,z)\sim(p_{data},Z)}[f_1(C(x) - C(G(z)))]$ $\mathcal{L}_{rel}^G(C) = \mathbb{E}_{(x,z)\sim(p_{data},Z)}[f_1(C(G(z)) - C(x))]$ | N/A | • This objective intends to measure how much realistic is a generated sample compared to a real one. • Any discriminator performing classification can be transformed into a relativistic counterpart. • Considers prior knowledge related to the data and the training scheme that other losses do not consider. |

encourage them to visit Arjovsky and Bottou's work [23] in which they discussed all these topics thoroughly and compare with examples the adversarial losses.

The GAN loss feature has 14 different losses in all the publications. It is possible to classify them into two groups: adversarial and auxiliary. Such losses intend to guide the architecture training towards the specific task and boost the general performance, respectively. Independent of the data domain and the computer vision task, most GANs use both adversarial and auxiliary losses in their implementation. In this case, however, 14 publications presented architectures that rely on both types of losses.

The most frequent auxiliary losses are the pixel-wise losses, *L2* [24–27] and *L1* [28–30] norm. Concerning the adversarial losses, the Vanilla GAN loss was the most popular present in 23 publications, followed by the WGAN variants. Fig 2C depicts the adversarial losses' distribution in the selected studies.

## Vanilla loss

The vanilla adversarial loss, defined by the cross-entropy loss, is implemented so that *G* aims to maximize it while *D* tries to minimize it. Interestingly, this loss approximates a function dependent on the Jensen Shannon Divergence ($\mathcal{D}_{JS}$) when *D* reaches its theoretical optimum.

Different from Kullback-Leibler divergence ($\mathcal{D}_{KL}$), a popular metric for distribution comparison, $\mathcal{D}_{JS}$ is a symmetric statistical distance, and its output bounds to [0, 1], an essential factor to stabilize neural network training. However, similar to $\mathcal{D}_{KL}$, $\mathcal{D}_{JS}$ cannot measure the distance between two distributions with disjoint supports. If, for instance, the real and generated distribution do not overlap, $\mathcal{D}_{JS} = 1$ independently of the distance between them, leading to a possible gradient vanishing during training.

From the 13 studies using vanilla adversarial loss, three publications used auxiliary losses, and another four used GAN augmentation as an intermediate state to boost the performance of a classification network; the rest focused their work on the augmentation process.

Most studies used traditional architectures such as CNN or vanilla generators and discriminators. Five used alternative architectures: StyleGAN [31–34], and ResNet [35]. And an additional work modified the generator architecture to produce annotated images [36]. Some publications presented pipelines that combine several GAN architectures [32,33,37–40] and a study that aims to compare the performance of different GAN architectures [34,41].

## Wasserstein adversarial loss

Given that conventional adversarial losses suffer from training instability by its intrinsic definition, Wasserstein GAN (WGAN) [42] introduces an adversarial loss robust against this. As its name implies, WGAN exploits the Wasserstein-1 distance as a loss function. The Wasserstein-1 distance can be elucidated as the minimum amount of work required to move an input distribution to match a target distribution. The main benefit of this measure is that its output is proportional to the distance between the distributions, independently of whether there are disjoint supports. However, its computational complexity makes its calculation infeasible with high-dimensional data.

Because of this, Arjovsky et al. proposed to use the Kantorovich-Rubinstein duality and reformulate the Wasserstein-1 distance into a maximization problem of K-Lipschitz functions [43]. The main challenge of this reformulation is to encourage the neural network architecture to represent K-Lipschitz functions during training. The first approach was to clip the network weights after each gradient update to lie in a compact space. Moreover, WGAN implementation replaces the discriminator with a critic *C* network that outputs a measure instead of a probability. Nevertheless, weight clipping adds a new hyperparameter to the training process

and limits the generator generalization power. As a consequence, Gulrajani et al. [44] encourages K-Lipschitz functions with a term addition to the objective function that penalizes the norm of the gradients of $C$ instead of limiting the weight values.

Wasserstein loss collects 19 studies when grouping regular (seven) and gradient penalty WGAN (twelve). Seven studies implemented auxiliary losses, 14 used the GANs as intermediate tasks (to boost classification, segmentation, and object detection), and twelve were implementations of already existing models without any modifications. There are studies using PGGAN [45,46], sinGAN [47], star-shaped GAN [48], and StyleGAN2 [28,30,49]. There is also a study presenting a new architecture with WGAN-GP loss [50] and an implementation of it from different authors [51,52].

## Relativistic adversarial loss

The Relativistic GAN (RGAN) is a modification to the GAN losses to measure how much more realistic a real sample is, compared to a generated one. A characteristic of GAN training is that $D$ is always fed with a pool of images, where half of them are generated and the other half are real. Since $D$ already knows the amount of real and synthetic data, the probability of classifying correctly a real image should decrease as the generated images start to get classified as real. A benefit of RGAN is that it can be easily applied to those adversarial losses using a regular discriminator. Analogous to WGAN, the discriminator can be seen as a critic with a final sigmoid layer; the critic measures how realistic an input is, while the activation function transforms that value into a probability. The authors used this new critic to subtract the output of real and generated images and then feed it into the activation layer. In this way, the output of the whole discriminator tells how much more realistic a sample is compared to another one. This implementation has independent objectives for the discriminator and generator that they train to maximize. Moreover, the generator now has an active role in both discriminator and generator objectives. Empirical results suggest that RGAN is more efficient in terms of performance, stability and complexity compared to Vanilla GAN, LSGAN and WGAN [53].

Two studies, which are the only ones employing relativistic GAN, are modifications of PathologyGAN, a model combining BigGAN, StyleGAN, and RGAN. In the first study, the authors included an "inverse generator" to transform images back into the input latent space and explore it [25]. On the other hand, the second study implemented an unmodified version of PathologyGAN and selected images based on their latent representation to boost a classification network [24].

## Conditional adversarial loss

Mirza and Osindero [54] extended the capacity of GANs with conditional GAN (cGAN). The idea of cGAN is simple yet powerful. It uses additional information $h$ that encourages $D$ to produce synthetic data containing the conditional information $h$. The benefit of cGAN is that $h$ can be any data type, including labels, vectors, or even images. Additionally, the great flexibility of cGAN allows any adversarial loss to be converted into conditional. One of the most popular conditional settings is to use an image as generator input. In this way, the GAN performs Image-to-Image (I2I) translation, a process where $G$ intends to transform images from an input domain into a target domain while preserving the meaning of the input image, e.g., transforming photographs into sketches. The most iconic architectures are Pix2Pix and CycleGAN.

This review retrieved six publications using conditional adversarial GANs. The first study used a 3D U-Net generator to transform synthetic segmentation masks into 3D fluorescence microscopy images [29]. The second study trained a conditional DCGAN and a diffusion

model with histology images to assess their output quality [15]. The third one implemented a conditional StyleGAN2 architecture to increase the performance in thyroid histopathology image classification [49]. The fourth publication designed a mix between a Variational Autoencoder (VAE) and GAN to generate blood cell images [55]. The fifth publication trained a conditional StyleGAN2 architecture to synthesize lung cancer cells and boost a classifier [56]. The last publication used Auxiliary Classifier GANs (ACGAN) [57] for federated learning in a hospital setup to produce histology images.

## Reproducibility and other features

17 publications shared their implementations, and 28 used public datasets to test their results (some of them also use their own datasets that are not publicly available). On the other hand, 23 publications had image augmentation as the main task of the study, 30 used unmodified GAN architectures published before, eleven performed an ablation study, and 26 did not use any baseline (Fig 2D). One could argue that since around 65% of the studies used unmodified GANs and 50% of publications focused on a different computer vision task, there is no need to do an ablation study, use a baseline, or share a code implementation. However, we believe that a reference point (baseline and ablation study) is essential to assess the performance of an approach that uses GANs, even for an intermediate task such as data augmentation to boost classification performance.

Regarding network architectures, regular CNN decoder and encoder are the most popular generator and discriminator architectures correspondingly, possibly because a significant part of the studies implemented unmodified GANs and that image augmentation was an intermediate task in the experimental setup. Interestingly, eight studies used variations of StyleGAN and three studies implemented PathologyGAN architecture. Another popular generator architecture was ResNet. Finally, the majority of architectures used an unsupervised training fashion; only eight implemented supervised and two self-supervised training.

## Measuring generative models performance

Measuring the performance of generative models has never been a trivial task. There are no ground truths, and realisticness is a subjective attribute that is highly biased depending on the evaluator. Moreover, linking such attributes to a quantitative value is even harder. Because of this, there has been extensive research on measuring generated data quality [58].

The current gold standard methods are based on neural networks, more specifically on the classification model Inception v3 [59]. The logic behind this idea is that a network must learn features that capture the quality of images to achieve good classification performance. Additionally, this network is pretrained with the ImageNet dataset [60]. A general purpose dataset with natural images from 100 classes.

The Inception Score (IS) is the first metric of this type [61]. It was designed under the following assumptions:

1. High-quality images produce classification outputs with low entropy. i.e., the output probability distribution of a single sample will tend to a single class.
2. A model with high generalization should produce a set of samples such that the marginal of the classification output tends to have high entropy. i.e., the average probability distribution of a large enough set of samples should tend to a uniform distribution (equal representation of all the classes).

Under these assumptions, a powerful generative model that produces high-quality and diverse samples will generate two distributions with low similarity. In this way, the IS of a generative model is defined as the exponential of the expectation of the $\mathcal{D}_{KL}$ between the inception conditional distribution and its marginal. The higher the IS, the more powerful the generative model.

One of the flaws of IS is that it relies solely on the generated data probability distribution, leaving behind the real data distribution. The Fréchet Inception Distance (FID) aims to tackle this by estimating the probability distributions of real and synthetic data from the intermediate features of Inception v3, which should extract meaningful attributes from the samples [62].

With this, the FID of the generated dataset is defined as the Wasserstein-2 distance between the probability distribution of the synthetic and real intermediate Inception features. Different from IS, a lower FID translates into a better sample quality. FID is currently the most popular metric in the computer vision field.

This systematic review collected more than 20 performance metrics, including qualitative assessment and metrics based on tasks different from image augmentation. Some recurrent metrics for image augmentation are IS and FID. The baseline feature is another with high variance, although a significant part of the studies did not use baselines to compare their results. There are at least 15 different baselines through the pool of publications. 10 studies also used baselines towards the principal task to solve, meaning that some baselines are not GAN architectures (refer to the baseline and performance metrics columns).

## Beyond image generation

The goal of this last subsection is to describe the approaches of the studies that implemented and used GANs with original strategies to make the most out of them.

Fluorescence microscopy is a technique focused on highlighting targeted proteins with fluorescent markers that emit light at a specific wavelength. This modality is used to study protein interaction, localization, expression and much more. However, there is a limited number of fluorescent proteins that can be used simultaneously due to overlapping absorption spectra, which makes it possible to run experiments with three to four targeted proteins solely. The approach of Osokin et al. [48] trained a star-shaped GAN to generate synthetic samples with multiple targeted proteins. They used each generator branch to produce each channel separately (i.e., each protein marker). The red channel (which targets the protein that defines the cell morphology) conditioned the rest of the channels. This is the first publication that our methodology retrieved, and it is the only study that focuses on generating images that are impossible to obtain in vivo.

## Sequential GANs

Another interesting approach to generating high-quality images is to use a pair of GANs sequentially. i.e., the first GAN is noise-to-image, while the second is an I2I translation architecture. We retrieved six studies that decided to implement sequential GANs for different reasons. Han et al. [38] generated annotated images by training the first GAN to produce segmentation masks from 3D noise inputs, enabling the generator to produce variable resolution masks as all the layers of it are convolutional; the second GAN takes the generated segmentation mask to produce the microscopy images. Moreover, The authors introduce a new discriminator configuration called multiscale discriminator, which uses several PatchGAN discriminators simultaneously to evaluate the image at different scales to gain insight from local and global features during training. Murali et al. [37] used a combination of DCGAN

and Enhanced super-resolution GAN (ESRGAN) [63]. It first produces low-resolution cell images and then increases their resolution with ESRGAN. This was the first study that used FID to evaluate the quality of the generated images. Barrera et al. [40] trained a WGAN to produce white blood cell images reproducing global features and an I2I translation GAN that adds the local (fine) features to the images to resemble real cell morphology. They trained a single WGAN to produce the initial images but trained an I2I GAN per blood cell type (five in total). Tang et al. [32] and Tasnadi et al. [33] used StyleGAN2 to produce segmentation masks to be used as the input of I2I GANs to produce 2D microscopy images. Finally, in a neutrophil study, the authors first generate single cells with a white marker in the cytoplasm that is later filled with blue-green death inclusions with an I2I GAN, similar to an image in painting setting [64].

## Generating annotated data

The sequential GAN fashion of Han et al., Tang et al., and Tasnadi et al. are not the only studies that generated annotated data. Dimitrakopoulos et al. [36] designed a generator with two sibling branches in the last three layers. It produces an RGB image and a binary mask simultaneously. It also uses a loss based on the Markov random field to smooth the outputs. It achieved the best IoU score using 20% of the real data. Instead of modifying the network architecture, Devan et al. [47] added color-coded cell bounding boxes to a limited dataset (10 samples) and trained a sinGAN [65] to produce images annotated with different cell types. After the image generation step, the coordinates of the bounding boxes are extracted and then deleted from the samples. Eschweiler et al. [29] is the only study included that did not implement a noise-to-image GAN because it first generates segmentation masks with simulation tools to then transform them into cell images with I2I translations. We decided to include this publication since it produces cell images from scratch.

## Discriminator applications

Continuing with the contribution of Han et al., there are three more studies that focused their effort on the discriminators instead of the image generation itself. The process of understanding whether an image is real or synthetic requires the discriminator to learn features that are of high importance for different tasks, giving it a potential that is usually overlooked. Kastaniotis et al. [27] sought to improve the quality of images by enforcing the discriminator to classify the image as real or synthetic plus generate a Class Activation Map (CAM) and encourage it to focus on critical regions. The authors extracted real image CAMs from a powerful pre-trained classification network and used a L2 norm to train the discriminator when the input was a real image. The full GAN architecture was able to produce synthetic images as well as weak object localization through the CAMs. Beyond the introduction of Info-WGANGP, Hu et al. [50] used the feature maps produced by the discriminator to do cell-level and image-level classification. Although the focus of this publication excluded the image generation quality, Anaam et al. [52] will later show that Info-WGANGP produces images capable of boosting classification network performance, surpassing DCGAN, WGAN, and WGAN-GP. Lastly, Rubin et al. [66] used transfer learning to fine-tune a discriminator. They first trained the GAN with unlabeled sperm cell images to transform it into a high-accuracy cancer cell classification network with limited labeled images, showing the potential of the features learned by the discriminators during training.

## Latent space exploration

Interestingly, GANs can generate images and find lower-dimensional embedded representations capturing important features in a smooth latent space. Besides Hu et al. [50] work, we retrieved four more studies that focus their efforts on exploiting the learned latent representations produced by the network. The first one was Quiros et al. [25], which trained a PathologyGAN and an auxiliary encoder with H&E stained cell images. The role of the encoder was to embed synthetic and real images back to the intermediate latent $W$ (StyleGAN latent space), which is disentangled. Moreover, this allowed the architecture to be further regularized by including reconstruction losses in such space. Latent $W$ was able to capture essential high-level features such as color, textures, and cell types in a logical and smooth distribution; it was possible to use this latent representation to cluster images with similar morphology without any sort of data annotation before the training.

Mascolini et al. [28] used the generation process as a pretext task to learn a self-supervised representation of cell images. They trained an unconditional StyleGAN architecture and extracted the intermediate features from the trained discriminator to assess classification performance in SVM. The latent space learned by StyleGAN was 5% less accurate than the latent space learned from trained classification networks. This is less than expected because StyleGAN did not require any type of data annotation, which increases the potential of latent representations. Dolezal et al. [56] used the latent space of a binary-conditioned StyleGAN2 trained with cancer histology images to identify the transition from healthy to cancer cells. The latent space was sufficiently smooth that linear interpolations between healthy and cancer cells showed the progression of morphological changes. Interestingly, the goal of exploring this latent space was to use the GAN generations to enhance the learning process of residents during educational sessions. The smooth interpolation allowed the students to identify subtle morphological changes and understand the disease progression more easily than without the generated samples. Lastly, Howard et al. [30] replaced the StyleGAN2 mapping network with a self-supervised learning decoder to produce a latent space that serves as generator input and contains meaningful features that can be used for model explainability and virtual biopsy to produce histology samples from MRI data.

## Weakly conditional generation

Besides Conditional GANs, there are alternative methods to control the features of generated images. This last section shows the publications that used unconditional GANs setups to control the generation process through different methodologies. Mirzazadeh et al. [45] trained a feature extractor and an unconditional GAN with rejection and non-rejection heart transplant histology images. Once both networks were trained, the feature extractor regularized the GAN input vectors, forcing it indirectly to produce heart transplant images that are underrepresented in the dataset. Reich et al. [31] designed a StyleGAN2-based architecture with sibling branches to produce two microscopy modalities simultaneously. Besides that, the authors fed the discriminator with real image sequences (ordered and shuffled) to indirectly enforce the generator to generate a pseudo-time series stack of images. Real ordered sequences should be classified as real, while the shuffled ones as synthetic. Giuste et al. [39] studied generated heart transplant rejection images with two different GANs. First, they trained a PGGAN with non-rejection images and then an Inspirational GAN for the rejection images using the latent space produced by PGGAN to locate the resembling vectors of rejection samples in the non-rejection latent space. The results indicated that non-rejection images tended to be located in the center of the Gaussian, while the rejection samples were on the border of it. The last study of this type was carried out by Ghose et al. [24], which trained a Pathology GAN with

a feature extractor to deal with data imbalance for breast cancer detection. The training was completely unconditional, but the latent space produced by the feature extractor guided the generative process to force the generator to produce diseased images exclusively. These images are later used for classification purposes.

Finally, Table 3 collects the included studies with the most relevant features extracted from the paper. A full version of the table is presented in S1 Table.

## Datasets

Table 4 arranges all the datasets cited by the pool of selected publications. This table gathers relevant metadata regarding the images, keynotes, examples of different applications for the dataset, and the URLs to download (all the provided URLs were last visited on 11/04/2025 for verification and accessibility). An extended version of the Cell microscopy dataset table, including the original and other applications, can be found in S2 Table.

### RxRx19a

The RxRx19a dataset [115] is the first compilation of cell images to study the morphological effects of COVID-19 in HRCE and Vero cells. It contains a pair of images and a 1,024-dimensional vector embedding extracted from a deep-learning approach trained with several cell types and perturbations. It has a total of 305520 fluorescence images of five channels corresponding to different stains. The stains are Hoechst, ConA, Phalloidin, Syto14, and WGA.

### smFISH dataset

This fluorescence dataset originates from a genetics study investigating the role of various genes in the nuclear RNA export pathway [111–113]. The authors used single-molecule FISH (smFISH) to track and quantify RNA molecules in gene knockout experiments conducted on MCF7 adenocarcinoma cells. Each fluorescence image contains four channels, highlighting different cellular components: PolyA$^+$ RNA (green), lncRNA NORAD (orange), the nucleus (blue), and nuclear speckles (red).

### RxRx1

The RxRx1 dataset [116] is a fluorescence microscopy dataset with robust experimental conditions to study biological variations with machine learning. It contains 125,510 6-channel images representing 1,108 different classes. Each stain highlights different cell organelles: nucleus, endoplasmic reticulum, actin cytoskeleton, nucleolus, mitochondria, and Golgi apparatus. The classes come from gene knockout experiments with small interfering RNA (siRNA), a nucleotide sequence that binds to the messenger RNA (mRNA) and impairs protein synthesis. In the experimental setup of the RxRx1 dataset, a single siRNA targets a single protein, and the 1108 classes correspond to the different siRNAs used by the researchers. Moreover, the experiments were carried out with four different cell types (HUVEC, RPE, U2OS, and HepG2).

### Human protein atlas image classification Kaggle competition

This dataset [127], composed of 31,072 fluorescence microscopy images, is presented by The Human Protein Atlas in a Kaggle competition to build an integrated tool to identify protein

**Table 3. Summary of the studies that met the eligibility criteria. Publicly available datasets are highlighted with bold text.**

| Publication | GAN loss | Generator | Discriminator | Performance metrics | Model baselines | Data type | Dataset |
|---|---|---|---|---|---|---|---|
| Wubineh et al. [67] (2025) | Vanilla GAN loss | Regular CNN decoder architecture | Regular CNN encoder architecture | Performance based in downstream task (Classification) | No baselines | Optical Microscopy | Private dataset |
| Preda et al. [18] (2025) | ACGAN [57] | Regular CNN decoder architecture | SwinT [68] | FID | Vanilla GAN, WGAN. **Discriminators:** ResNet [69], ViT [70] | Optical Microscopy | **TCGA-CRC-DX** [71] |
| Barrera et al. [64] (2024) | WGAN, Vanilla GAN | Regular CNN decoder architecture, U-Net | Regular CNN encoder architecture, Regular CNN encoder architecture | Performance based on downstream task (Classification) | No baselines | Optical Microscopy | **Barrera et al. dataset** [72] |
| Nayar et al. [73] (2024) | Vanilla GAN | Regular CNN decoder architecture | Regular CNN encoder architecture | Performance based on downstream task (Classification) | No baselines | Optical Microscopy | **Liquid based-cytology Pap smear dataset** [74] |
| Khan et al. [75] (2024) | Vanilla GAN | Regular CNN decoder architecture | Regular CNN encoder architecture | Performance based on downstream task (Classification) | No baselines | Optical Microscopy | **PBC dataset** [76], **LISC dataset** [77], **Raabin WBC** [78] |
| Ngasa et al. [79] (2024) | WGAN-GP | Regular CNN decoder architecture | Regular CNN encoder architecture | FID, IS | WGAN-GP, Improved Denoising Diffusion Probabilistic Models (DDPM) [80] | Optical Microscopy | Ngasa et al. Dataset [79] |
| Niehues et al. [34] (2024) | Vanilla GAN (with gradient penalty) | StyleGAN2 | | FID | Latent diffusion model (LDM) [81], KL-decoder LDM, VQ-decoder LDM | Optical Microscopy | **NCT-CRC-HE-100K dataset** [82] |
| Van Booven et al. [17] (2024) | Vanilla GAN | Regular CNN decoder architecture | Regular CNN encoder architecture | IS | cGAN, StyleGAN [83] | Optical Microscopy | **The Cancer Genome Atlas (TCGA)**, Private dataset (available under request) |
| Howard et al. [30] (2024) | WGAN-GP, L1 (in W space) | StyleGAN2 | | Performance based on downstream task (Latent space exploration, reconstruction, feature preservation) | Encoder4Editing [84] | Optical Microscopy | **TCGA**, **Clinical Proteomic Tumor Analysis Consortium (CPTAC)** |
| Khan et al. [35] (2023) | Vanilla GAN | ResNet | ResNet | Performance based on downstream task (Classification) | No baselines | Optical Microscopy | Private dataset, AI-Hub, **Liquid based-cytology Pap smear dataset**, **SIPaKMeD dataset** [85] |
| Singh et al. [86] (2023) | Vanilla GAN | Regular CNN decoder architecture | Regular CNN encoder architecture | Performance based on downstream task (Classification) | No baselines | Optical Microscopy | **BreakHis dataset** [87] |
| Dee et al. [49] (2023) | Conditional WGAN-GP | StyleGAN2-ADA | | FID | No baselines | Optical Microscopy | Tharun and Thompson dataset (available under request) [88], **Niki-TCGA dataset** [89], **TCGA** |
| Giuste et al. [39] (2023) | WGAN-GP, Inspirational GAN (IGAN) [90] | Progressive growing GAN (PGGAN) [91] | | Performance based on downstream task (Classification) | DDPM | Optical Microscopy | Private dataset |
| Ghose et al. [24] (2023) | RGAN, L2 | PathologyGAN | | Performance based on downstream task (Classification) | No baselines | Optical Microscopy | Private dataset (available under request) |

(*Continued*)

**Table 3**. (Continued)

| | | | | | | | |
|---|---|---|---|---|---|---|---|
| Barrera et al. [40] (2022) | WGAN, Vanilla GAN | Regular CNN decoder architecture, U-Net | Regular CNN encoder architecture, Regular CNN encoder architecture | FID, IS,Learned Perceptual Image Patch Similarity (LPIPS) | No baselines | Optical Microscopy | Private dataset |
| Kunzmann et al. [15] (2022) | Conditional Vanilla GAN | Vanilla GAN | | Qualitative assessment | LDM | Optical Microscopy | **Asthma Equidae dataset** [92] |
| Dolezal et al. [56] (2022) | Conditional WGAN | StyleGAN2 | | FID, qualitative assessment | No baselines | Optical Microscopy | **TCGA, CPTAC** |
| Rando et al. [93] (2022) | Vanilla GAN | Regular CNN decoder architecture | Regular CNN encoder architecture | FID, IS | No baselines | Optical Microscopy | Private dataset |
| Pandya et al. [94] (2022) | Vanilla GAN | Regular CNN decoder architecture | Regular CNN encoder architecture | No metrics | No baselines | Optical Microscopy | Leukemia dataset |
| Liu et al. [55] (2021) | Conditional Vanilla GAN | SGAN [95] | | IS | DCGAN [19], ProGAN [96] | Optical Microscopy | **BCCD** [97] |
| Yu et al. [98] (2021) | Vanilla GAN | Regular CNN decoder architecture | Regular CNN encoder architecture | Performance based on downstream task (Classification) | No baselines | Optical Microscopy | Private dataset |
| Zhao et al. [99] (2021) | WGAN | Regular CNN decoder architecture | Regular CNN encoder architecture | FID | No GAN baselines | Optical Microscopy | **Zheng et al.** [100], Private dataset |
| Mirzazadeh et al. [45] (2021) | WGAN-GP | PGGAN | | Mattthews correlation coefficient (MCC) in the classifier | No baselines | Optical Microscopy | **DNA-based transplant rejection** [101], **Children's hospital of Atlanta** [102–105] |
| Quiros et al. [25] (2020) | RGAN, L2 | PathologyGAN | | Qualitative assessment | No baselines | Optical Microscopy | **Netherlands Cancer Institute** [106], **Vancouver General Hospital databases** [107] |
| Murali et al. [37] (2020) | Vanilla GAN | Regular CNN decoder architecture | Regular CNN encoder architecture | FID, qualitative assessment | No baselines | Optical Microscopy | Private dataset |
| Teramoto et al. [46] (2020) | WGAN | PGGAN | | Accuracy, Sensitivity, Specificity (downstream task) | DCGAN | Optical Microscopy | Private dataset |
| Almezh-ghwi et al. [108] (2020) | Vanilla GAN | Regular CNN decoder architecture | Regular CNN encoder architecture | Performance based on downstream task (Classification) | No baselines | Optical Microscopy | **LISC dataset** |
| Chen et al. [26] (2020) | WGAN-GP, L2 | ResNet | Regular CNN encoder architecture | FID | Vanilla GAN | Optical Microscopy | **Pap-smear dataset** [109] |
| Wang et al. [16] (2019) | WGAN | WGAN | | Qualitative assessment | No baselines | Optical Microscopy | Private dataset |
| Hu et al. [50] (2018) | WGAN-GP, Mutual information (MI) | ResNet | ResNet | Performance based on downstream tasks (Segmentation, classification and clustering) | No baselines | Optical Microscopy | **BM dataset** [110], Private dataset |
| Tang et al. [32] (2024) | Vanilla GAN (with gradient penalty) | StyleGAN2 | | FID | DCGAN, PGGAN | Fluorescence microscopy | **smFISH dataset** [111–113] |
| Mascolini et al. [28] (2022) | WGAN, Jacobian Regularization, L1, R1 regularization [114] | StyleGAN2 | | Accuracy | WGAN, Pre-trained CNN (ImageNet) | Fluorescence microscopy | **RxRx19a Sars-CoV-2 image collection** [115], **RxRx1 dataset** [116] |

(*Continued*)

**Table 3**. (Continued)

| Eschweiler et al. [29] (2021) | cGAN, L1 | 3D U-Net [117] | PatchGAN | normalized root mean squared error (NRMSE), structural similarity index measure (SSIM), Zero mean normalized cross-correlation (ZNCC) | No baselines | Fluorescence microscopy | **Willis et al. [118], Faure et al. [119]** |
|---|---|---|---|---|---|---|---|
| Reich et al. [31] (2021) | Vanilla GAN, R1 regularization, Path length [120] | Two parallel convolutional paths (StyleGAN generator) for each domain style | U-Net with adaptive discriminator augmentation | IS, FID, Fréchet Video distance | StyleGAN2, StyleGAN2 3D | Time-lapse fluorescence microscopy | Private dataset |
| Tasnadi et al. [33] (2023) | Vanilla GAN (with gradient penalty) | StyleGAN2 | | FID | No baselines | Immunofluorescence microscopy | **Salivary Gland Tumor and Fallopian datasets [121]** |
| Anaam et al. [122] (2023) | WGAN-GP, MI | ResNet | ResNet | FID | DCGAN, WGAN, WGAN-GP, InfoGAN [123] | Immunofluorescence microscopy | **I3A dataset [124]** |
| Anaam et al. [51] (2023) | WGAN-GP, MI | ResNet | ResNet | FID | No baselines | Immunofluorescence microscopy | **I3A dataset** |
| Anaam et al. [52] (2021) | WGAN-GP, MI | ResNet | Two output layers CNN | FID, Classifier two-sample test (C2ST) | DCGAN, WGAN, WGAN-GP | Immunofluorescence microscopy | **I3A dataset** |
| Dimitrakopoulos et al. [36] (2020) | Vanilla GAN, Markov Random Field-based loss | Regular CNN decoder architecture | Regular CNN encoder architecture | FID, IoU in auxiliary segmentation models trained with real and generated data | Vanilla GAN, No GAN-based augmentation | Fluorescence microscopy, Optical microscopy | **BBBC038v1 dataset [125]** |
| Verma et al. [126] (2020) | Vanilla GAN | Regular CNN decoder architecture | Regular CNN encoder architecture | Performance based on downstream task (Classification) | No baselines | Fluorescence microscopy | **Human Protein Atlas Image Classification Kaggle competition [127]** |
| Hussain et al. [41] (2020) | Vanilla GAN | Regular CNN decoder architecture | Regular CNN encoder architecture | Classification Accuracy, Kolmogorov–Smirnov (KS) distance of the first principal component of features extracted by additional feature extraction network | VanillaGAN, ProGAN | Widefield fluorescence microscopy | Private dataset |
| Kastaniotis et al. [27] (2018) | Vanilla GAN, L2 | Regular CNN decoder architecture | Regular CNN encoder with attention map generation | Qualitative assessment | No baselines | Fluorescence microscopy | **Hep-2 Cells Classification contest [128]** |
| Osokin et al. [48] (2017) | WGAN-GP | Star-shaped generator | Regular CNN encoder architecture | C2ST | Vanilla GAN, WGAN | Fluorescence microscopy | **LIN dataset [129]** |
| Devan et al. [47] (2021) | WGAN-GP | sinGAN [65] | | Performance based on downstream task (Object detection) | No baselines | Transmission electron microscopy | Private dataset (available under request) |
| Han et al. [38] (2018) | Vanilla GAN | Regular CNN decoder architecture | Multiscale PatchGAN | mean IoU, Average cell size, average mitochondria size and roundness, average number of mitochondrias per cell, realisticness (SVM real, synthetic classifier) | Zhao et al. Pipeline [130], Non-parametric beseline, DCGAN | Transmission Electron Microscopy | **VNC dataset [131]** |
| Rubin et al. [66] (2018) | Vanilla GAN | Regular CNN decoder architecture | Regular CNN encoder architecture | Sensitivity, Specificity, AUC (for the classification task) | No GAN baselines (metrics based on classification only) | interferometric phase microscopy | Private dataset |

**Table 4. Cell microscopy datasets used in the studies meeting the eligibility criteria.**

| Dataset | Modality | Number of samples | Resolution | Annotated | Cell type | URL |
|---|---|---|---|---|---|---|
| RxRx19a Sars-CoV-2 image collection [115] (2020) | Fluorescence microscopy | 305,520 | $1,024 \times 1,024 \times 5$ | No | HRCE, Vero | https://www.rxrx.ai/rxrx19 |
| smFISH dataset [111–113] (2020) | Fluorescence microscopy | 99 | $1,024 \times 1,024$ | Yes | MCF7 | https://data.mendeley.com/datasets/6hsf4fyhsn/2 https://data.mendeley.com/datasets/9s9m4wytfw/1 https://data.mendeley.com/datasets/cv7n2bbcb4/1 |
| RxRx1 dataset [116] (2019) | Fluorescence microscopy | 125,510 | $512 \times 512 \times 6$ | Yes | HUVEC, RPE, U2OS, HepG2 | https://www.rxrx.ai/rxrx1 |
| Human protein atlas image classification Kaggle competition [127] (2019) | Fluorescence microscopy | 31,072 | $512 \times 512$ or mix between $2048 \times 2048$ and $3072 \times 3072$ | Yes | Human cell | https://www.kaggle.com/competitions/human-protein-atlas-image-classification/ |
| LIN dataset [129] (2017) | Fluorescence microscopy | 170,000 | $80 \times 48$ or $160 \times 96$ | Yes | Yeast | https://github.com/aosokin/biogans |
| Salivary Gland Tumor and Fallopian datasets [121] (2023) | Immunofluorescence microscopy | 40 | Variable resolution | Yes | Fallopian tube and Salivary gland tumor | https://zenodo.org/records/8096773 |
| I3A dataset [124] (2016) | Immunofluorescence microscopy | 13,596 | Unspecified | Yes | Hep-2 cell line | https://hep2.unisa.it/dbtools.html |
| *Arabidopsis thaliana* dataset [118] (2016) | Confocal fluorescence microscopy | 125 | Between $326 \times 367 \times 107$ and $512 \times 512 \times 396$ | Yes | *Arabidopsis thaliana* | https://www.repository.cam.ac.uk/handle/1810/262530 |
| BioEmergences datasets [119] (2016) | Fluorescence microscopy | 394 | $512 \times 512$ with slices from 104 to 120 | No | *Danio rerio* | http://bioemergences.iscpif.fr/bioemergences/openworkflow-datasets.php |
| Hep-2 Cells Classification Contest [128] (2013) | Fluorescence microscopy | 28 | $1,388 \times 1,038$ | Yes | Hep-2 cell line | https://mivia.unisa.it/datasets/biomedical-image-datasets/hep2-image-dataset/ |
| BBBC038v1 [125] (2012) | Fluorescence microscopy, Optical microscopy | 670 | From $256 \times 256$ to $1,040 \times 1,388$ | Yes | Different organisms including human, mice and flies | https://bbbc.broadinstitute.org/BBBC038 |
| Barrera et al. dataset [72] (2024) | Optical microscopy | 5,605 | $360 \times 363$ | Yes | neutrophils | https://data.mendeley.com/datasets/rh3jw43hjs/1 |
| Ghose et al. dataset [24] (2023) | Optical microscopy | Unspecified | $7,000 \times 6,000$ approx. | Yes | Human cell | Accessible under request |
| Raabin WBC [78] (2022) | Optical microscopy | 20,936 | Unspecified | Yes | Blood cells | https://raabindata.com/free-data/ |
| Asthma Equidae dataset [92] (2021) | Optical microscopy | 6 | Unspecified | Partially | Equine blood cells | Accessible under request |
| Tharun and Thompson dataset [88] (2021) | Optical microscopy | 156 | Unspecified | Yes | Thyroid gland tumor cells | Available under request to sekretariat.patho@uksh.de |
| PBC dataset [76] (2020) | Optical microscopy | 17,092 | $360 \times 363$ | Yes | Peripheral Blood cells | https://data.mendeley.com/datasets/snkd93bnjr/1 |
| Liquid-based cytology Pap smear dataset [74] (2019) | Optical microscopy | 963 | $2,048 \times 1,536$ | Yes | Skin cancer | https://data.mendeley.com/datasets/zddtpgzv63/4 |
| SIPaKMeD dataset [85] (2018) | Optical microscopy | 4,049 | $2,048 \times 1,536$ | Yes | Pap-smear | https://www.cs.uoi.gr/~marina/sipakmed.html |
| NCT-CRC-HE-100K dataset [82] (2018) | Optical microscopy | 100,000 | $224 \times 224$ | Yes | Human colorectal cancer | https://zenodo.org/records/1214456 |
| BCCD dataset [97] (2017) | Optical microscopy | 365 | $640 \times 480$ | Yes | Blood cells | https://github.com/Shenggan/BCCD_Dataset |
| BreakHis dataset [87] (2016) | Optical microscopy | 9,109 | Variable resolution | Yes | Breast cancer cells | https://web.inf.ufpr.br/vri/databases/breast-cancer-histopathological-database-breakhis/ |

(*Continued*)

**Table 4**. (Continued)

| | | | | | | |
|---|---|---|---|---|---|---|
| Nikiforov dataset (2016) | Optical microscopy | 278 | Variable resolution | Yes | Encapsulated follicular variant papillary thyroid carcinoma | https://image.upmc.edu/NikiForov%20EFV%20Study/view.apml |
| BM dataset [110] (2015) | Optical microscopy | 11 | $1,200 \times 1,200$ | Yes | Healthy human bone marrow | https://github.com/pkainz/MICCAI2015/ |
| Vancouver General Hospital cohort [107] (2011) | Optical microscopy | 1,286 | $1,128 \times 720$ | Yes | Breast cancer cells | https://tma.im/tma_portal/C-Path/supp.html |
| Pap-smear image dataset [109] (2005) | Optical microscopy | 917 | Variable resolution (single-cell dataset) | Yes | Pap-smear | http://mde-lab.aegean.gr/index.php/downloads |
| HCMV dataset [47] (2021) | Transmission electron microscopy | 350 | $2,048 \times 2,048$ | Yes | *Herpesviridae*, human *cytomegalovirus* | Accessible under request |
| VNC dataset [131] (2013) | Transmission Electron Microscopy | 40 | $1,024 \times 1,024$ | Yes | *Drosophila melanogaster* instar larva ventral nerve cord | https://github.com/unidesigner/groundtruth-drosophila-vnc/tree/master |
| TCGA-CRC-DX [71] | Optical Microscopy | 51,918 | $512 \times 512$ | Yes | Colorectal Cancer | https://zenodo.org/records/3832231 |

locations from high-throughput images. Beyond the expected high classification accuracy, the organizers also included hardware limitations to encourage efficiency and reproducibility.

Users can freely download the dataset on the webpage in two versions: a scaled set of $512 \times 512$ images and a high-resolution version of $2,048 \times 2,048$ and $3,072 \times 3,072$ images. This dataset contains 27 cell types with different morphology, and each image is annotated with 28 protein organelle localization labels. From the 4-channel images, contestants were expected to use the green channel to predict the labels, while the other three as references.

## LIN dataset

The LIN dataset [129] compiles 170,000 fluorescence images of fission yeast, highlighting two types of proteins. The red channel signals the Bgs4 protein, which localizes the areas of cells' active growth, while the green channel signals the so-called polarity factors that mark areas of the cells' cortex. These polarity factors define cell geometry. Each image in the LIN dataset corresponds to a single cell, and the dataset was created to study the interactions between the polarity factors. The dataset targets 41 different polarity factors that control cellular polarity in different ways.

## Salivary gland tumor and fallopian datasets

This dataset was obtained from two patients: a salivary gland tumor from a 29-year-old male and fallopian tube tissue from a 64-year-old female [121]. The entire dataset was manually annotated for segmentation, comprising a total of 2,876 cells in high-resolution images. Tasnadi et al. [33] extracted $256 \times 256$ effective tiles for their experiments.

## I3A dataset

This fluorescence dataset [124] has been used in different classification challenges, including ICPR 2012, ICIP 2013, and ICPR 2014. It contains HEp-2 cells with a total of 68,429 images (13,596 test images and 54,833 training images). The images were automatically segmented with the DAPI channel and manually annotated by experts. The images are labeled with 6 different classes (homogeneous, centromere, speckled, nucleolar, mitotic spindle, and Golgi), and each sample also has metadata including label, cell intensity, cell mask, and ID. This dataset is only available under request.

### *Arabidopsis thaliana* dataset

The *A. thaliana* dataset [118] is the product of a publication carried out by Willis et al. where they studied the size and growth regulations of this plant with a Python pipeline. To do so, the authors designed an *A. thaliana* strain with fluorescence markers targeting different genes. There are three channels; the authors used the yellow and red channels to study the cell membranes and nucleus, and the green channel was not analyzed in the study. Each sample corresponds to a stack of volumetric data, which is automatically segmented and partly curated. It has a total of 125 3D stacks (of variable resolution) of six different *A. thaliana* apical meristem.

### BioEmergences datasets

The BioEmergences group is a member of the France-BioImaging National Infrastructure that develops methodologies and tools for the observation, quantification, and modeling of biological processes. They published a workflow [119] to reconstruct cell lineage using 3-dimensional time series fluorescence datasets of stained embryos from Zebra fish (*Danio rerio*), ascidian (*Phallusia mammillata*), and sea urchin (*Paracentrotus lividus*). Each dataset is composed of stain nuclei and membranes $512 \times 512$ stacks with variable 3rd dimension resolution.

Eschweiler et al. [29] only used the Zebra fish data to train their GAN architecture, since they also used the *Arabidopsis thaliana* dataset.

### Hep-2 cells classification contest

The MIVIA HEp-2 Images dataset [128] was used in the Cell classification contest carried out in the ICPR 2012. It is an Indirect ImmunoFluorescence (IIF) dataset composed of 28 40× images of $1,388 \times 1,038$ manually labeled and segmented. Each image has metadata including the label (positive, negative, or intermediate intensity), number of objects, number of cells, and number of mitoses.

### BBBC038v1

Kaggle 2018 Data Science Bowl compiled the BBBC038v1 dataset [125] from different laboratories to encourage the development of nuclei segmentation models. It has a total of 841 2D light microscopy images with different staining protocols to increase the variability in the dataset. The whole dataset comprises a total of 37,333 segmented nuclei of different cell types and techniques due to the sampling protocol.

### Barrera et al. dataset

This optical microscopy dataset contains over 5,000 neutrophil images from peripheral blood smears [72]. It is annotated for image classification, with each sample classified into one of seven categories: Normal neutrophils (NEU), Hypogranulated neutrophils (HYP), neutrophils containing cryoglobulins (CRY), Döhle bodies (DB), Howell-Jolly body-like inclusions (HJBLI), green-blue inclusions of death (GBI), and phagocytosed bacteria (BAC).

### Ghose et al. dataset

This H&E tissue microarray (TMA) dataset is used by Ghose et al. in their image augmentation study [24]. The main characteristic of this dataset is that all samples come from patients with a history of breast cancer event (BCE). The authors extracted the samples from the

pathology databases of Oxford University (training cohort) and the Singapore General Hospital (validation cohort). From 133 cases in total, the training cohort (67) comprises one to three TMA cores per patient, while the validation cohort (66) includes three TMA cores per patient. Each H&E TMA core image is approximately $7,000 \times 6,000$ pixels.

### Raabin WBC

The Raabin Health Database is an initiative designed to enhance data accessibility for artificial intelligence specialists and medical informatics professionals [78]. The white blood cell (WBC) dataset samples are fully annotated for cell classification into neutrophils, eosinophils, basophils, lymphocytes, and monocytes, while a random subset of the samples is also annotated for segmentation. Each annotation step was carried out by expert biologists.

### Asthma equidate dataset

The samples of this dataset [92] come from Equine bronchoalveolar lavage fluid. The cells were cytocentrifugated and stained with May-Grundwalb Giemsa and then digitalized. Of the six slides, two images are fully annotated, and four are partially annotated. A veterinary pathologist carried out the whole annotation process.

### Tharun and Thompson dataset

This H&E stained dataset [88] is collected from the pathology archives at the University Clinic Schleswig-Holstein and the Woodland Hills Medical Center. Each whole slide image corresponds to a stained section per tumor taken at 40× magnification. Each sample was annotated according to the agreement between two pathologists. The whole dataset has five classes: follicular thyroid carcinoma, follicular thyroid adenoma, noninvasive follicular thyroid neoplasm with papillary-like nuclear features, follicular variant papillary thyroid carcinoma, and classical papillary thyroid carcinoma.

### PBC dataset

This dataset contains over 17,000 manually annotated images of individual white blood cells for classification [76]. Each image is categorized into one of eight classes: neutrophils, eosinophils, basophils, lymphocytes, monocytes, immature granulocytes, erythroblasts, and platelets (thrombocytes). Note that each image captures a single white blood cell, but red blood cells may also be present.

### Liquid-based cytology Pap smear dataset

This dataset consists of pap smear images from 460 patients obtained using the liquid-based cytology technique. It contains 963 images, classified into four categories of pre-cancerous and cancerous cervical lesions. The dataset was collected to support computer-assisted tools aimed at accelerating diagnosis.

### SIPaKMeD dataset

This dataset consists of manually cropped, isolated cells from Pap smear slides, designed for studying normal and pathological cervical cells using computer vision tools. The images are categorized into five classes: superficial-intermediate cells, parabasal cells, koilocytotic cells, dyskeratotic cells, and metaplastic cells. The SIPaKMeD dataset contains a total of 4,049 images.

### NCT-CRC-HE-100K dataset

The NCT-CRC-HE-100K dataset is an H&E-stained histology dataset that compiles samples from healthy and colorectal cancer tissues. It contains 100,000 images representing nine different tissue types: adipose (ADI), background (BACK), debris (DEB), lymphocytes (LYM), mucus (MUC), smooth muscle (MUS), normal colon mucosa (NORM), cancer-associated stroma (STR), and colorectal adenocarcinoma epithelium (TUM). The repository provides different data splits for various applications.

### BCCD

The BCCD dataset [97] is a blood cell dataset designed to train cell detection models. It has three labels (red cell, white cell, and platelet), and it is composed of 365 light microscopy $640\times 480$ images. The images are available in a GitHub repository, where there are data preparation scripts for abnormality recognition.

### BreakHis dataset

The Breast Cancer Histopathological Image Classification (BreakHis) dataset contains histology images of breast tumor tissue from 82 patients at various magnification levels. The images are categorized as benign or malignant, and the dataset is periodically updated. Within these two main categories, the images are further classified by tumor type, resulting in four benign and four malignant tumor types.

### Nikiforov dataset

The samples of this encapsulated follicular variant of papillary thyroid carcinoma (EFVPTC) dataset are taken from 210 patients of different ages diagnosed with noninvasive and invasive EFVPTC. The whole dataset is a collection from 13 different sites in 5 countries. Moreover, each sample was manually curated by 24 thyroid pathologists from 7 countries. The classes in this dataset are: EFVPTC, invasive follicular variant papillary thyroid carcinoma, classical papillary thyroid carcinoma, follicular thyroid carcinoma, and benign.

### BM dataset

The BM dataset [110] is an annotated histology dataset consisting of eleven $1,200\times 1,200$ images of human bone marrow. Each image has annotated nuclei, resulting in 4205 detected nuclei. The authors of this dataset developed a cell detection model.

### NIK and VGH cohorts

This H&E-stained histological dataset is a merge of the Netherlands Cancer Institute and Vancouver General Hospital cohorts [107]. The images were sampled from 576 patients, and 1,286 of them related to breast cancer. The authors of this dataset produced a prognostic model by extracting significant features to assess the prognosis from microscopic image data.

### Pap-smear image dataset

This dataset [109] has 917 images of human cells stained with the Papanicolau method. Each sample is a single cell with manual annotation inside 7 labels, and it additionally has 20 related numerical features. The authors published this dataset with the idea of a benchmark dataset for classification methods.

### HCMV

The Herpes Human Cytomegalovirus (HCMV) dataset [47] contains 350 transmission electron microscopy (TEM) $2,048 \times 2,048$ images of human skin fibroblasts infected with HCMV. Each image has bounding boxes annotated by experts. Each box was classified with the three HCMV capsid envelopment stages: naked, budding, and enveloped. Although this dataset is not publicly available, the authors are open to sharing it upon request.

### VNC dataset

The VNC dataset [131] is composed of two stacks of 20 serial section transmission electron microscopy sections of the *Drosophila Melanogaster* third instar larva ventral nerve cord. One stack has segmentation annotations highlighting neuron membranes, mitochondria, synapses, and glia/extracellular space.

### TCGA-CRC-DX

This dataset is a subset of the TCGA database and compiles colorectal cancer histology microscopy images. After preprocessing, the images are normalized and have $512 \times 512$ resolution. Beyond the image data, the samples are annotated based on their Microsatellite Instable (MSI) status with the labels high (MSI-H), low (MSI-L), and none (MSS). The annotation of each image corresponds to the patient's status.

### Risk of bias with ROBIS

Following the guidelines of ROBIS, we present the concern and rationale for each domain proposed there. Additionally, S3 Table compiles the complete ROBIS implementation.

### Concerns regarding specification of study eligibility criteria

**Low Concern**. All signaling questions were answered as "Yes" or "Probably Yes", so no potential concerns about the specification of eligibility criteria were identified.

### Concerns regarding methods used to identify and/or select studies

**High Concern**. Some eligible studies are likely missing from the review, as no additional searching was conducted beyond databases. Moreover, only one person was responsible for screening titles, abstracts, and full texts to classify a study as eligible. Finally, papers written exclusively in English were considered.

### Concerns regarding methods used to collect data and appraise studies

**High Concern**. Some bias may have been introduced since only one person was responsible for data collection, and the nature of the studies does not allow a suitable risk of bias assessment.

### Concerns regarding the synthesis and findings

**High Concern**. The synthesis is likely to produce biased results because it was not possible to consider bias between studies, and the nature of the studies does not allow accounting for variation between them.

 **Risk of bias in the systematic review: High.** The main source of risk of bias is that a single person was responsible for the whole review process. However, there is another source of bias,

and it is the nature of the studies. To our knowledge, there are no available tools or methodologies to assess the risk of bias in DL studies, and not all the ROBIS criteria fit in a systematic review in this field.

## Discussion

### Considerations on model implementation

**The GAN implementation heavily depends on the study goal**. Although the initial thought is that any generative model must produce images with high-quality scores in terms of image quality metrics, this review showed that this might not always be the case. Image quality metrics such as FID or IS do not necessarily correlate with the performance of other computer vision tasks like classification or object detection [18]. As mentioned earlier in the Reproducibility and other features section, over 40% of the studies used GANs to indirectly boost the performance of downstream tasks and did not always evaluate image generation quality explicitly.

We retrieved two studies whose primary goal was to train GANs to boost the performance of classification networks that reported high FID scores, which translate into low-quality images according to the metric [37,99]. However, the classification networks presented a significant increase in accuracy when trained with the generated data. In contrast, there are studies in which image quality is crucial, for instance when images cannot be manually sampled [48] or when the generated data is intended to be used as an educational tool [56].

Our reasoning is that tasks such as classification rely mostly on global features, and therefore, the classification models could tend to ignore the local features. Considering that classification was the most recurrent goal among the retrieved downstream tasks, one could justify why vanilla GAN loss is still relevant even in 2025 when there are more sophisticated losses. Nevertheless, we encourage the researchers to always take realism into consideration because the main applications of medical imaging require high fidelity due to their ethical concerns and possible effects in case of wrong decisions.

We observed from the studies that GANs could generate high-quality images but lacked in reproducing fine details [34,37,46], an essential property in biology. Fortunately, this review showed how the authors used the flexibility of GANs to deal with those limitations with different approaches beyond the adversarial loss. We found pipelines of sequential GANs, auxiliary networks, discriminator modifications, novel auxiliary losses, transfer learning, and training single GANs per class.

**GANs offer more than just synthetic images**. We saw in the Discriminator applications section how the adversarial training process produces additional outputs that can be easily overlooked because of the direct application of GANs. One of those products is the discriminator. It is common that after training, the whole focus is given to the generator, and the discriminator is discarded. However, the studies collected here proved that discriminators can have as much value as the generator thanks to the features it learns from the data. A well-trained discriminator is capable of learning features for unsupervised classification [50], transfer learning [66], and CAMs [27] for a wide range of applications.

Moreover, the Latent space exploration section showed the potential of the generator intermediate features. Studies showed how the latent space produced by an unconditional generator can be employed to add constraints to the generation process without supervision [56] or to make annotations and samples simultaneously [25]. The possibilities increase when auxiliary architectures are part of the entire pipeline, to the point of performing virtual biopsy simulation with meaningful latent features [30].

**With all its potential, GANs are still a challenging model**. The past decade of study on GANs has revealed that they possess distinct limitations by definition [132]. First, they are well known for their training stability; keeping a good balance between the generator and discriminator is mandatory to preserve stable gradients, and they are highly sensible to hyperparameters as well. Second, an intrinsic limitation of these models is mode collapse [133], i.e., when the model cannot learn all the modes of the data distribution and ends up generating samples with limited variability.

The research in image generation has continued to the point that GANs are no longer considered state-of-the-art anymore. Diffusion models [134] replaced GANs in the last years due to their ability to better capture data distribution, more training stability, and generation flexibility. Unfortunately, diffusion models require significantly more resources than GANs to generalize and generate samples as well. Although it is a new approach, there are already review papers on medical image generation with them [135].

There are already studies comparing the performance of GANs and diffusion models in cell imaging [15,34,39]. However, the amount is still limited, and the information is insufficient to conclude. Even with their drawbacks, GANs have unique qualities that diffusion models cannot offer, such as latent space exploration, easy accessibility in terms of resources, or meaningful intermediate features from both generator and discriminator. Even more, **the adversarial concept is still relevant in computer vision**, to the point that there is active research on the potential of merging features of GANs and diffusion models [136]. Moreover, this review showed their relevance in the last years with a constant publishing rate and how they have enough capacity to meet the requirements of researchers to boost the performance of downstream tasks with accessible resources in reasonable time frames.

Finally, implementing image augmentation with GANs must be carried out carefully to minimize the risk of bias during training. First, a model should always be trained with both real and generated images, since GANs are likely to not reproduce the whole data distribution because of their mode collapse [49,51]. Second, mixing classical augmentation methods (flip, rotation, patches) and generative augmentation [45] showed the best improvement in downstream tasks in the studies retrieved here. And third, preserving a good balance between real and synthetic samples is essential in augmentation [40,51]. The results suggest that performance improvement is not always proportional to the amount of generated data used for augmentation during training [52].

## Challenges, limitations, and opportunities

### Experimental design

**The experimental design is one of the biggest flaws that we encountered in the eligible studies**. It is not possible to reproduce, compare, and assess the models presented in the selected studies. The Reproducibility and other features section showed that each model is trained and tested with different datasets, each study employs different baselines with distinct performance metrics, and a significant part of them do not share their source code implementation. Even though data augmentation is generally not the main task of the studies, we believe that relying solely on downstream tasks like classification or object detection to assess generative performance should be considered carefully.

Such downstream tasks are likely to be performed by other ANNs which are susceptible to bias during training. Even more, researchers suggest that pretraining models with general-purpose datasets can affect the features a network focuses on to decide in the medical domain. Yu et al. show in their study how a classification network pretrained on ImageNet tends to give attention mostly to the cell nucleus, while a pretrained network in a cell dataset focused

on the whole cell organelles [98]. Supporting the evaluation of image augmentation with additional measurements will give better insight into the image quality and, therefore, ensure adequate performance of downstream architectures.

## Performance metrics, evaluation, and realism

**Evaluate and comparing GAN studies is currently not feasible**. Another limitation we perceived in this study is how to evaluate GANs performance. One of the reasons for this difficulty is the performance metrics; this study shows how there are different metrics, and although some can be transformed into others, the high number of options limits the assessment between models.

**Measuring realism is an intrinsic challenge of generative modeling**. Tasks like I2I translation and image augmentation heavily rely on producing realistic images. If it is not possible to accurately measure the generated data quality, it will be impossible to compare models or estimate their generalization.

**Medical and microscopy imaging require more robust metrics than other fields**. Some of the most popular metrics, such as IS and FID, are based on ANNs trained with a general-purpose dataset, ImageNet [60], raising the concern of whether such metrics can capture and evaluate the features describing the quality of images that are likely to be significantly different from daily life objects as mentioned earlier in this discussion. A potential alternative to this problem is to train the feature extractor with a cell microscopy benchmark dataset to ensure high accuracy and precision in the FID and IS metrics.

**It is essential to focus attention on how results are measured**. Tronchin et al. [137] assess and propose a framework for GANs in the context of medical imaging. A good starting point could be to use the work of Borji [138] as a reference, where he presents and evaluates different qualitative and quantitative performance metrics for GANs.

Considering the mentioned aspects, **the joint efforts between biology and DL experts are of vital importance**. Developing good practices such as including ablation studies, baseline comparisons, and defining gold standards for metrics and datasets can only be established with continuous collaboration between the experts of both fields. Only biology experts know the nature and applications of the data, while DL experts know how to design experimental setups together with how to interpret and evaluate the outputs from the models.

## Ethical considerations and risk of bias

Although the real applications of generative models are increasing significantly in the last few years, there is a lack of discussion about the ethical implications of using them in the medical field [139]. The power of these tools has increased to the point that synthetic images have been used for scams or cyberattacks [140]. Even more, the data alone carries its ethical considerations since it contains sensible information from the patients.

Sun et al. divided VAEs and GANs into five core components and analyzed the possible attacks and defense approach separately, providing insight on how to build robust pipelines against cyberattacks [141]. On the other hand, Ning et al. evaluate the ethical implications of generative artificial intelligence based on the nine ethical principles in healthcare context, resulting in a comprehensive checklist of practices to ensure ethical guidelines in the field [139].

Finally, **the risk of bias in this review could be high**, given that a single person was responsible for conducting the search, screening, and analysis of studies, and considering that we retrieved studies solely from five databases. However, we must remember that the

PRISMA guidelines were not designed for systematic reviews in the field of DL, and some criteria are not applicable. The systematic review is a practice that should be extended to other fields beyond health, but an adaptation is required to make it possible.

### Good practices for GANs in image augmentation

We aim to conclude the discussion with a brief list of good practices to assist researchers interested in utilizing generative models for image augmentation. The information presented here comprises our own findings from this review, as well as results and conclusions drawn from the authors of the studies analyzed in this work.

- Avoid solely training with generated samples [47,52].
- Utilize both classical and generative augmentation techniques [45,52,108].
- Leverage diverse large datasets for fine-tuning [66].
- Explore using the discriminator features for downstream tasks [27,50,66].
- Carefully assess pretrained models from different domains [46,98].
- Implement common quality metrics for reproducibility.
- Evaluate image quality and downstream task performance separately.
- Adapt network configuration for limited data. e.g., multiscale discriminator [38] or sinGAN [65].
- Pipelines of sequential GANs could be useful when recovering local features is difficult [37,38,40,64].
- Consider training a single GAN per class when dealing with highly variable datasets [40, 51].
- Leverage intermediate features from generator for soft-conditional generation [24,31,39].
- Design experiments and pipelines always considering ethical implications and discuss them explicitly [139].

### Conclusion

This publication is, to our knowledge, the first systematic review of GANs with cell microscopy for image augmentation. In this study, we examine, summarize, and discuss the most popular methods, together with the current trends in how researchers approach problems in the domain of cell microscopy imaging with GANs. We also compiled and compared the public cell microscopy datasets that have been used for image augmentation since the introduction of GANs to give a brief overview of the available options. Finally, we identified some inconsistencies related to the experimental set-up that should be considered not only in studies involving GANs but also in any generative modeling study. Reproducibility and comparability are essential to speeding up the research progress in any field. For that reason, we propose a list of good practices to alleviate this matter.

### Supporting information

**S1 Table. Summary of the studies that met the eligibility criteria, full version.**
(PDF)

**S2 Table. Cell microscopy datasets used in the studies meeting the eligibility criteria, full version.**
(PDF)

**S3 Table. Risk of bias with ROBIS.** Complete implementation.
(PDF)

**S4 Table. List of all the studies retrieved with the search strategy.**
(PDF)

**S5 Table. PRISMA checklist.**
(PDF)

## Acknowledgements

We thank Miro Mirada (DFKI) for useful discussion and comments on the manuscript.

## Author contributions

**Conceptualization:** Duway Nicolas Lesmes-Leon, Sheraz Ahmed.

**Funding acquisition:** Andreas Dengel.

**Investigation:** Duway Nicolas Lesmes-Leon.

**Supervision:** Andreas Dengel, Sheraz Ahmed.

**Writing – original draft:** Duway Nicolas Lesmes-Leon.

**Writing – review & editing:** Duway Nicolas Lesmes-Leon, Andreas Dengel, Sheraz Ahmed.

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
