## [Decision Letter · Decision Letter 0]

PONE-D-23-26522Generative adversarial networks in cell microscopy for image augmentation. A systematic reviewPLOS ONE

Dear Dr. Lesmes-Leon,

Thank you for submitting your manuscript to PLOS ONE. After careful consideration, we feel that it has merit but does not fully meet PLOS ONE’s publication criteria as it currently stands. Therefore, we invite you to submit a revised version of the manuscript that addresses the points raised during the review process.

We look forward to receiving your revised manuscript.

Kind regards,

Abel C.H. Chen

Academic Editor

PLOS ONE

Journal Requirements:

 "This study was partially funded by Sartorius AI Lab (SAIL), a collaboration between Sartorius (https://www.sartorius.com) and the German center for artificial intelligence (DFKI) (https://www.dfki.de). The funders

had no role in study design, data collection and

analysis, decision to publish, or preparation of the

manuscript."

Please provide an amended statement that declares *all* the funding or sources of support (whether external or internal to your organization) received during this study, as detailed online in our guide for authors at http://journals.plos.org/plosone/s/submit-now.

Please also include the statement “There was no additional external funding received for this study.” in your updated Funding Statement.

"This study was partially funded by Sartorius AI Lab (SAIL), a collaboration between Sartorius (https://www.sartorius.com) and the German center for artificial intelligence (DFKI) (https://www.dfki.de). The funders

had no role in study design, data collection and

analysis, decision to publish, or preparation of the

manuscript."

We note that one or more of the authors is affiliated with the funding organization, indicating the funder may have had some role in the design, data collection, analysis or preparation of your manuscript for publication; in other words, the funder played an indirect role through the participation of the co-authors. If the funding organization did not play a role in the study design, data collection and analysis, decision to publish, or preparation of the manuscript and only provided financial support in the form of authors' salaries and/or research materials, please do the following:

1.) Review your statements relating to the author contributions, and ensure you have specifically and accurately indicated the role(s) that these authors had in your study. These amendments should be made in the online form.

2.) Confirm in your cover letter that you agree with the following statement, and we will change the online submission form on your behalf: 

Reviewers' comments:

Reviewer's Responses to Questions

**Comments to the Author**

1. Is the manuscript technically sound, and do the data support the conclusions?

Reviewer #1: Partly

Reviewer #2: Partly

2. Has the statistical analysis been performed appropriately and rigorously? 

Reviewer #1: Yes

Reviewer #2: N/A

3. Have the authors made all data underlying the findings in their manuscript fully available?

Reviewer #1: Yes

Reviewer #2: Yes

4. Is the manuscript presented in an intelligible fashion and written in standard English?

Reviewer #1: Yes

Reviewer #2: Yes

5. Review Comments to the Author

Reviewer #1: This article is the first systematic review of cellular GANs and summarizes and discusses the most popular methods as well as describes how to use them, which is instructive overall, but for the small number of experiments mentioned in the article that fail to reproduce the structure, this part of the experiment should probably not be ignored. And I don't think it's a very full workload for a review.

Reviewer #2: This paper provides a review on GAN in cell microscopy for image augmentation. It is considered as a review articles which is not technical enough to suit the requirement to be published in PLOS One. Furthermore, as a review article, the analysis provided is a bit difficult to follow. Authors should provide a critical review instead of extensive descriptions. A more conclusive/novel analysis need to be done to be the main contribution of the paper.

6. PLOS authors have the option to publish the peer review history of their article (what does this mean?). If published, this will include your full peer review and any attached files.

Reviewer #1: No

Reviewer #2: No

---

## [Author Response · Author response to Decision Letter 1]

We first would like to thank all the reviewers that took the time to read carefully the manuscript and gave their corrections for the sake of improving the quality of the present work. Here we will point out the changes made in the text to solve the concerns arisen during the first revision.

Reviewer #1: This article is the first systematic review of cellular GANs and summarizes and discusses the most popular methods as well as describes how to use them, which is instructive overall, but for the small number of experiments mentioned in the article that fail to reproduce the structure, this part of the experiment should probably not be ignored. And I don't think it's a very full workload for a review.

Our Response: Thank you very much for your insight. We decided to extend the results and discussion sections to give a more detailed insight of the studies that met the selection criteria in the review. We also included a section of good practices for GANs in image augmentation that readers can use as a starting point and minimize possible difficulties that can arise during model implementations. Furthermore, we offer these suggestions based on the results of the retrieved studies.

Unfortunately, it was not clear for us what the reviewer referred as “experiments that fail to reproduce the structure” considering that this is a systematic review and there is no experiments section. We, however, make a discussion about the experimental designs that authors normally use in their studies. This information is presented in Table 3, Table S1, and we discussed the limitations of the current practices.

Reviewer #2: This paper provides a review on GAN in cell microscopy for image augmentation. It is considered as a review articles which is not technical enough to suit the requirement to be published in PLOS One. Furthermore, as a review article, the analysis provided is a bit difficult to follow. Authors should provide a critical review instead of extensive descriptions. A more conclusive/novel analysis need to be done to be the main contribution of the paper.

Our Response: Thank you very much for your comments, we revisited our manuscript and included the following changes according to your insights:

1. Although PLOS ONE is a journal who does not accept regular review papers for publication, it does accept systematic reviews that fulfill the requirements to be considered as such. In this manuscript, we applied and followed the PRISMA 2020 statement, which are the guidelines to write systematic reviews. We would like to point out that the PRISMA statement was designed for systematics reviews in the field of health, where there are specific experimental designs that do not fit the main domain of this manuscript (deep learning). We implemented as much as possible the PRISMA 2020 statement, considering that GAN augmentation in cell microscopy has a direct impact in the health and medicine domain. Moreover, we discussed the limitations of the current statement when the main topic of the systematic review deviates from the health domain.

2. We re-arranged our manuscript to facilitate the reading. This included moving the technical definitions into the results sections and removing some information that we considered could confuse the readers.

3. We added the “Beyond image generation” subsection in the results to increase the discussion and assess the potential of GANs. We described how GANs can be implemented in different ways to achieve different goals beyond only generating synthetic samples. This addition also helped us to make a more in-depth discussion into what are the different alternatives to implementing a GAN, depending on the nature of the problem.

4. Finally, we now presented a list of good practices with the aim of helping the researchers that would like to implement GANs in the cell microscopy domain. This list seeks to increase the experiment's reproducibility and guide the research in the implementation process. Each of the points of this list come from the results and conclusions of the studies that we reviewed in this work.

---

## [Decision Letter · Decision Letter 1]

PONE-D-23-26522R1Systematic review of generative adversarial networks in cell microscopy: trends, practices, and impact on image augmentationPLOS ONE

Dear Dr. Lesmes-Leon,

Thank you for submitting your manuscript to PLOS ONE. After careful consideration, we feel that it has merit but does not fully meet PLOS ONE’s publication criteria as it currently stands. Therefore, we invite you to submit a revised version of the manuscript that addresses the points raised during the review process.

We look forward to receiving your revised manuscript.

Kind regards,

Abel C.H. Chen

Academic Editor

PLOS ONE

Journal Requirements:

Reviewers' comments:

Reviewer's Responses to Questions

**Comments to the Author**

1. If the authors have adequately addressed your comments raised in a previous round of review and you feel that this manuscript is now acceptable for publication, you may indicate that here to bypass the “Comments to the Author” section, enter your conflict of interest statement in the “Confidential to Editor” section, and submit your "Accept" recommendation.

Reviewer #1: All comments have been addressed

2. Is the manuscript technically sound, and do the data support the conclusions?

Reviewer #1: Partly

3. Has the statistical analysis been performed appropriately and rigorously? 

Reviewer #1: Yes

4. Have the authors made all data underlying the findings in their manuscript fully available?

Reviewer #1: No

5. Is the manuscript presented in an intelligible fashion and written in standard English?

Reviewer #1: Yes

6. Review Comments to the Author

Reviewer #1: The authors should describe the characteristics of the microscopic cell image data. What are the challenges in terms of processing methods compared to general image data?

7. PLOS authors have the option to publish the peer review history of their article (what does this mean?). If published, this will include your full peer review and any attached files.

Reviewer #1: **Yes: **Haigen Hu

---

## [Author Response · Author response to Decision Letter 2]

We first would like to thank all the reviewers that took the time to read carefully the manuscript and gave their corrections for the sake of improving the quality of the present work. Here we will point out the changes made in the text to solve the concerns arisen during the first revision.

Journal Requirements: Please review your reference list to ensure that it is complete and correct. If you have cited papers that have been retracted, please include the rationale for doing so in the manuscript text, or remove these references and replace them with relevant current references. Any changes to the reference list should be mentioned in the rebuttal letter that accompanies your revised manuscript. If you need to cite a retracted article, indicate the article’s retracted status in the References list and also include a citation and full reference for the retraction notice.

Our Response: We revisited every single citation and searched it in the Retraction Watch Database to be completely sure that we were no citing retracted publications. This allowed us to update some preprints that are already published. We only found a single publication with a publisher correction that is exclusively related to the copyright license (https://doi.org/10.1038/s41592-019-0612-7), and one dataset URL link that is not available anymore. For this last citation, we decided to remove it and explain the situation in the manuscript.

1. Have the authors made all data underlying the findings in their manuscript fully available?

he PLOS Data policy requires authors to make all data underlying the findings described in their manuscript fully available without restriction, with rare exception (please refer to the Data Availability Statement in the manuscript PDF file). The data should be provided as part of the manuscript or its supporting information, or deposited to a public repository. For example, in addition to summary statistics, the data points behind means, medians and variance measures should be available. If there are restrictions on publicly sharing data—e.g. participant privacy or use of data from a third party—those must be specified.

Reviewer #1: No

Our Response: As mentioned earlier, we discovered that the URL of the Leukemia dataset is currently inaccessible. This was a dataset published in Kaggle, but for some reason is not there anymore. We removed this citation and updated the manuscript according to that. Other than that, every dataset and publication are correctly cited and can be accessed freely. Every comment and conclusion made in our manuscript is exclusively based on the publications and datasets.

Reviewer #1: The authors should describe the characteristics of the microscopic cell image data. What are the challenges in terms of processing methods compared to general image data?

Our Response: Thank you very much for your insight. We followed your comments and extended the introduction section, where we discussed the unique challenges that cell microscopy imaging has. We did not modify neither results nor discussion sections, since we already mentioned and discussed those works that directly addressed some concerns added in the introduction.

---

## [Decision Letter · Decision Letter 2]

PONE-D-23-26522R2Systematic review of generative adversarial networks (GANs) in cell microscopy: trends, practices, and impact on image augmentationPLOS ONE

Dear Dr. Lesmes-Leon,

Thank you for submitting your manuscript to PLOS ONE. After careful consideration, we feel that it has merit but does not fully meet PLOS ONE’s publication criteria as it currently stands. Therefore, we invite you to submit a revised version of the manuscript that addresses the points raised during the review process.

We look forward to receiving your revised manuscript.

Kind regards,

Abel C.H. Chen

Academic Editor

PLOS ONE

Reviewers' comments:

Reviewer's Responses to Questions

**Comments to the Author**

1. If the authors have adequately addressed your comments raised in a previous round of review and you feel that this manuscript is now acceptable for publication, you may indicate that here to bypass the “Comments to the Author” section, enter your conflict of interest statement in the “Confidential to Editor” section, and submit your "Accept" recommendation.

Reviewer #3: (No Response)

2. Is the manuscript technically sound, and do the data support the conclusions?

Reviewer #3: Partly

3. Has the statistical analysis been performed appropriately and rigorously? 

Reviewer #3: I Don't Know

4. Have the authors made all data underlying the findings in their manuscript fully available?

Reviewer #3: Yes

5. Is the manuscript presented in an intelligible fashion and written in standard English?

Reviewer #3: No

6. Review Comments to the Author

Reviewer #3: Generative Adversarial Networks (GANs) have received a lot of attention and have seen significant improvements since their introduction over a decade ago. However, applying GANs to specific domains, as discussed in this manuscript, present greater challenges.

The objective of the current manuscript is to show the usage of GANs for cell microscopy data augmentation and to provide insights into it. Additionally, the manuscript includes a contribution in the form of a list and discussion regarding the datasets. These may encourage researchers to explore novel methodologies using such databases, fostering interdisciplinary projects between biologists and DL specialists.

Although the goals have been set, the manuscript requires further polishing.

While reading the Adversarial Losses section, I noticed that the authors used a lot of mathematical terms without proper definitions, explanations, or references. For instance, what are Z, X, and X’, mathematically speaking? Further, for a non-mathematician audience, this section may be difficult to follow. This made me wonder about the target audience that the authors had in mind. As the authors, in Lines 206-207, recommend other studies for formal definitions, I believe that they should try to provide a more straightforward and intuitive explanation of Adversarial Losses, using graphical interpretations whenever possible.

When explaining the GANs (lines 184-202), it should also be better to mention how G and D improve their performance along the training process, whenever they fail. This detail could also be included in Fig 3. Besides, I suggest remaking Fig 3, because it could be more informative and appealing. Moreover, I expected to see diagrams for each architecture of the selected studies.

Quality metrics such as ID and FID are not elucidated upon their initial introduction, nor are their limitations for use in cell microscopy images properly addressed. It could be a really nice contribution. Such a discussion is only shortly presented towards the conclusion of the manuscript (lines 740-744).

As the authors are presenting a systematic review, I anticipated clearly defined research questions. However, the presentation consists of four bullet points, which are commented on throughout the manuscript. I would have expected a format comprising questions and corresponding answers (but, in fact I do not know if it is mandatory for a S.R.).

I agree that it is relevant to consider preprints such as those in ArXiv and bioRxiv repositories. Specifically for the present manuscript, both repositories retrieved 4 studies for the final list (if I am not wrong). Thus, the authors not only must emphasize they are preprints but also evaluate their qualities (by running the source codes/experiments, checking data, etc.). Further, when filling out the ROBIS tool, is it possible to take into account the issue of preprint repositories (unpublished reports)?

It seems that you might not provide precise definitions or citations for Vanilla GAN and CNN in your study. I am not sure whether the Vanilla GAN you are referring to is the same as Goodfellow's GAN, and whether it uses Convolutional Neural Networks or not. It is important to ensure that your selected studies are not using Vanilla and CNN-based GANs interchangeably, as this could lead to confusion in your arguments. I am concerned about the inconsistencies in your statements on lines 170-173 and 656-659 and the number of Vanilla GANs mentioned in your study.

The discussion section has interesting parts. However, I expected that all of them would be anchored with findings and statements along with the manuscript. Further, statements cannot arise out of the blue. For instance, Line 751: The joint efforts between biology and DL experts are of vital importance. Ok, perhaps it is common sense, but it was not clearly tackled in your manuscript.

If the list of good practices was gathered from the studies, for each item, I believe the authors need to cite them and make some comments, exemplifying the use of each item, in order to be better grounded.

I believe that, whenever we discuss the problem of image generation for biological/medical/chemical/etc sciences, a discussion about image generation ethics must be presented. There are many hot discussions about it (e.g., https://www.nature.com/articles/d41586-024-00372-6). Thus, a few words can be necessary.

Line 8. “, it allows”. Replace by “. It allows” (or which allows)

Reference [5] is incomplete. I recommend checking all reference standards.

Line 63: “Although some of these reviews compile some microscopy”. Remove the second “some”.

Line 98: Check this statement (“neither has nor”).

Line 169: Citations are missing: “to the introduction of Pix2Pix [???] and CycleGAN [???] in 2017”.

Line 204: “2014 [4], several loss variants, architectures”. Replace by “2014 [4]. Several loss variants, architectures”.

Lines 205-206: “Table 2 summarizes the main characteristics of adversarial losses.”. Replace with “Table 2 summarizes the main characteristics of popular adversarial losses.” Since it seems it is not an exhaustive list and the authors argue in the caption they are popular adversarial losses (not all of them). In fact, is it popular for your cell microscopy or for all area studies? Check if the use of popular is the best choice.

Lines 209-2010: What are the differences between classes and groups?

Line 210: “two big groups, adversarial and auxiliary”. Suggestion “two groups: adversarial and auxiliary”.

Line 211: “pretend to guide”. I believe the authors would like to say “intend to guide". Check all occurrences of “pretend(s)”. I suppose you are making a mistake of false cognate between“pretende” from Spanish/Portuguese.

Observe that you do not follow a standard in equations in Table 2 (second column).The min/max on the left side, the sup on the right, or absence of min/max/sup/inf in equations.

Line 243: “the authors of Wasserstein GAN designed an adversarial loss robust against this instability.”. Suggestion “Wasserstein GAN (WGAN) [XXXX] has an adversarial loss robust against this instability”.

You repeat “only” many times in the manuscript. I believe you can remove almost all of them.

Line 214: “only 11 publications.” You have to cite them, to follow a standard, because in other places you cite. The same for lines 231 and 232. Please check through the manuscript whenever similar statements occur.

Standardize : baseline or base-line?

Line 335: Provide the complete name and cite FID.

Line 337: “Some studies also”. “Some” is not precise.

Line 376: “simultaneously, it also used a”. Replace by “simultaneously. It also uses a” (use of dot and check the verb tense.)

Line 410: “Besides Hu et al. work”. Cite it.

The dataset link of VNC dataset [99] (2013) in Table 4 seems not correct. The link directs to an image rather than a dataset.

https://mivia.unisa.it/datasets/biomedical-image-datasets/hep2-image-dataset/ At the moment I reviewed it, it was down.

https://hep2.unisa.it/dbtools.html. At the moment I reviewed it, it was down.

Line 414. Rewrite to avoid double W: “W. W”.

Line 485: “cite the Human Protein Atlas in a Kaggle competition”. Cite all datasets you explain in this section to follow a standard.

Line 523: “In one of their publications, [87], BioEmergences published”. Suggestion ”BioEmergences published [87]”

Line 665: It seems that this statement is incomplete. You observed and you concluded that... ?

Lines 678-681: include citations to the claims.

Lines 683-687: include citations to the claims.

Line 691: Here you discourage the reader of your study to continue using GANs for your challenges. Besides, it is a strong statement that DM replaced GANs. DMs, indeed, have gained much more attention than GANs in the last years, but GANs have their characteristics that have been explored (see https://arxiv.org/abs/2112.07804).

Line 714: “legible studies”. Replace it by “eligible studies”

Line 717: “a significant part do not share a code implementation”. Suggestion “a significant part of them do not share their source codes”.

Figures 1-3 are presented in low quality. I do not know if it was an issue with the submission system. Anyways, I recommend to the authors, if the journal allows, provide vector-based images (such as svg) in order that the figures do not lose their resolution.

7. PLOS authors have the option to publish the peer review history of their article (what does this mean?). If published, this will include your full peer review and any attached files.

Reviewer #3: No

---

## [Author Response · Author response to Decision Letter 3]

We first would like to thank the reviewer that took the time to read carefully the manuscript and gave their corrections for the sake of improving the quality of the present work. Here we will point out the changes made in the text to solve the concerns arisen during the first revision.

Reviewer #3: Generative Adversarial Networks (GANs) have received a lot of attention and have seen significant improvements since their introduction over a decade ago. However, applying GANs to specific domains, as discussed in this manuscript, present greater challenges.

Our Response: Thank you very much for all the detailed comments, this was an enormous help to improve not only the writing but also the content of our manuscript. We revised and grouped all of them, so there is no need to mention one by one.

• Grammar and spelling: We Corrected all the pointed grammar and spelling typos. This also includes reference inclusion within the text, and possible ambiguities from the manuscript’s older version.

• References and URL links: We revisited and updated all the references to avoid missing fields. We also double-checked the datasets URL and confirmed that all URL links were working by the visit time. Moreover, we decided to include in both text and references the last date of access to avoid any ambiguity.

• “As the authors are presenting a systematic review, I anticipated clearly defined research questions. However, the presentation consists of four bullet points, which are commented on throughout the manuscript. I would have expected a format comprising questions and corresponding answers (butin fact, I do not know if it is mandatory for a S.R.)”: We followed in the PRISMA statement which was designed specifically for systematic reviews. It is worth mentioning that such a protocol was designed having different fields in mind (such as psychology or medicine) and the field of computer science does not fit perfectly in all the requirements. In this case, the protocol requests the reviewers to either define research questions or objectives, and we decided to use objectives that are achieved throughout the manuscript in different ways (text, tables and good practices). This information is available in our PRISMA 2020 checklist supplementary table.

• Preprints, their validity, and possible bias source: There were indeed 4 publications that stayed as preprint from the 12 retrieved in both arXiv and bioRxiv. We went through the publications again and included a brief section in the text explaining that only one study published their implementation, but their architecture is an unmodified StyleGAN2-ADA, one of the most robust GANs, which achieved state-of-the-art results when it was first published. Moreover, there are more studies collected in this review implementing StyleGAN variations and achieving high scores, discarding the possibility of failure in the cell microscopy domain.

• Vanilla GAN and CNN GAN: We included a short paragraph explaining the definition of Vanilla GAN and how it can be used to represent loss and architectures independently to avoid possible misunderstandings. In the case of line 170-173, we did found a single study that implemented Vanilla GAN (loss and architecture), and even though it was published after 2016, the authors from that study concluded the lack of performance of GAN compared to a different baseline. On the other hand, our statements from lines 656-659 referred exclusively to the vanilla adversarial loss, so we adjust the paragraph to avoid ambiguities.

• Citations with group of studies: As we make a several study clustering during the results to easily explain them, we decided to follow the next criteria to make in-text citation.

1. Studies that have information that is not easily retrieved from the Table 3.

2. Study clusters easily retrieved from Table 3 with a size smaller than five to avoid repetitive citing

3. Single studies that we considered to be worth mentioning (as an exemplification or by information that is unavailable in Table 3).

• Table 2: We updated the caption of table two to “Summary of adversarial losses”. In this case, only the Hinge adversarial loss is left to complete all the adversarial losses, but we decided not to include it since we did not find a publication that implemented it.

We also updated the equation notation in the case of least-square loss. The min/max terms are specified only when there is ambiguity, such as on Vanilla loss where one network seeks to maximize and the other to minimize it, since a loss term is most times meant to be minimized. WGAN loss has the supremum term to the right of the equality since it is on terms of f, a term that does not directly influence W. All these equations are extracted from their original publications.

• Discussion section: We now linked each discussion section to its respective result section and cite the studies that exemplify the statements to facilitate the reading process. We also re-wrote ambiguous segments that lead to misunderstanding (including lines 665 and 751 from the older manuscript version).

Re re-organized the information where we mention diffusion models, our first thought was always to show that although Diffusion is gaining attention, GANs are still competitive with different advantages over Diffusion. We expect that this section facilitates the understanding of this.

• List of good practices: We clarified that each point in this list can either be a conclusion from our review or from the authors of the selected studies (can be implicit conclusion). Any claim that comes from a study is correctly cited.

• Adversarial losses section: Here we explicitly defined Z, X, and X’, and also cited accordingly the definition. We also briefly extended the definition to facilitate the understanding of this section. However, we decided not to reduce the mathematical concepts explained here because this is the core idea behind GANs, and they are the essentials to understand other GAN publications. Our idea is to give the reader the most important concepts so they can explore the field even beyond the papers we discussed in our review. We also updated the Figure 3 to facilitate the GANs definition with real examples.

On the other side, our first draft included detailed dynamics of the data distribution during training based on the adversarial loss, but we decided to exclude them from the manuscript because then the mathematical complexity raises significantly. This is why we invited those readers interested in the math to go to Arjovsky and Bottou's work, where they make a formal and detailed dissection of the adversarial losses.

Finally, we did not include detailed architecture figures because the core domain blocks (as in figure 3) are the most recurrent. Including layer-level architecture information and figures was out of the scope of this review.

• Image quality metrics: We included a short description of both the Inception Score and Fréchet Inception distance in the results section that will help the reader better understand the discussion part where we talk about the limitations of them in the field of cell microscopy.

• Ethical implications: We included a section within the discussion about the ethical considerations of generative models with medical imaging. We exemplified use cases, and also reference guidelines to meet ethical standards.

• Image quality: We followed all the formatting guidelines for images and supplementary information, including PACE as the PLOS ONE tool to ensure image quality. We believe that the manuscript PDF reduces the resolution of images, but a higher resolution counterpart is stores in the submission separately.

---

## [Decision Letter · Decision Letter 3]

PONE-D-23-26522R3Systematic review of generative adversarial networks (GANs) in cell microscopy: trends, practices, and impact on image augmentationPLOS ONE

Dear Dr. Lesmes-Leon,

Thank you for submitting your manuscript to PLOS ONE. After careful consideration, we feel that it has merit but does not fully meet PLOS ONE’s publication criteria as it currently stands. Therefore, we invite you to submit a revised version of the manuscript that addresses the points raised during the review process.

We look forward to receiving your revised manuscript.

Kind regards,

Abel C. H. Chen

Academic Editor

PLOS ONE

Reviewers' comments:

Reviewer's Responses to Questions

**Comments to the Author**

1. If the authors have adequately addressed your comments raised in a previous round of review and you feel that this manuscript is now acceptable for publication, you may indicate that here to bypass the “Comments to the Author” section, enter your conflict of interest statement in the “Confidential to Editor” section, and submit your "Accept" recommendation.

Reviewer #4: All comments have been addressed

2. Is the manuscript technically sound, and do the data support the conclusions?

Reviewer #4: Partly

3. Has the statistical analysis been performed appropriately and rigorously? 

Reviewer #4: N/A

4. Have the authors made all data underlying the findings in their manuscript fully available?

Reviewer #4: Yes

5. Is the manuscript presented in an intelligible fashion and written in standard English?

Reviewer #4: Yes

6. Review Comments to the Author

Reviewer #4: My comments are listed below:

- Since this is a systematic review, all databases containing preprints (such as arXiv and bioRxiv) should be excluded, as the research should focus solely on high-quality papers that have already undergone a peer-review process.

- Additionally, the review is limited to studies published until May 2023 (likely due to the paper revisions), but several works have been published in the past year and a half. Therefore, the trends and challenges described in the paper may have changed or should be updated.

- The authors should also provide a more thorough justification for why they chose to analyze only one type of generative model (GAN) for a specific task (augmentation), especially since such models are widely used today, particularly for domain translation (synthetic staining) or image enhancement/standardization (staining normalization).

7. PLOS authors have the option to publish the peer review history of their article (what does this mean?). If published, this will include your full peer review and any attached files.

Reviewer #4: No

---

## [Author Response · Author response to Decision Letter 4]

We first would like to thank the reviewer that took the time to read carefully the manuscript and gave their corrections for the sake of improving the quality of the present work. Here we will point out the changes made in the text to solve the concerns arisen during the first revision.

From a general perspective, we revisited the manuscript and corrected possible grammar mistakes and typos from our last rebuttal, as well as updated data from the new included studies.

1) Since this is a systematic review, all databases containing preprints (such as arXiv and bioRxiv) should be excluded, as the research should focus solely on high-quality papers that have already undergone a peer-review process.

Our Response: We sincerely appreciate your feedback and took it into careful consideration. While we previously defended the inclusion of preprints, we re-evaluated this decision in light of your comments and ultimately chose to retain them in our study for the following reasons:

▪ The PRISMA 2020 statement emphasizes transparency and completeness rather than restricting inclusion to peer-reviewed studies. Excluding preprint databases may compromise the completeness of the review, particularly in fast-moving research areas where recent developments have not yet undergone peer review.

▪ Our goal was not only to synthesize existing findings but also to analyze common practices in the application of GANs to microscopy. We believe that including preprints offers a more comprehensive and less biased view of current practices — both good and bad — in this multidisciplinary space where collaboration between Computer Science and Biology is essential.

▪ As noted in lines 167–169, only 4 out of the 15 preprints have not been peer-reviewed, and notably, the most recent two were submitted in 2024 and 2025. Excluding preprints would have resulted in the omission of up to 11 peer-reviewed studies, representing approximately 25% of all included studies. This highlights how preprint databases can help identify peer-reviewed work that might otherwise be missed due to indexing delays.

2) Additionally, the review is limited to studies published until May 2023 (likely due to the paper revisions), but several works have been published in the past year and a half. Therefore, the trends and challenges described in the paper may have changed or should be updated.

Our Response: We completely agree with your observation. The version you reviewed was submitted in April 2024, and while we are not aware of the reasons behind the delay in the review process, we recognize that timeliness is especially important for systematic reviews.

To address this, we updated our search to include studies published up to April 2025, identifying 14 additional eligible studies. We observed that GANs remain actively used in the field, with a steady publication rate over the past years. While not all of the new studies used public datasets, we noticed a growing trend toward their adoption, which we find encouraging.

Importantly, the main trends and patterns we identified in the 2024 version of our review still persist. The inclusion of new studies did not substantially change the statistics derived from our binary feature analysis. Therefore, our core insights remain valid in 2025, and we have updated the manuscript where relevant to reflect the new data.

3) The authors should also provide a more thorough justification for why they chose to analyze only one type of generative model (GAN) for a specific task (augmentation), especially since such models are widely used today, particularly for domain translation (synthetic staining) or image enhancement/standardization (staining normalization).

Our Response: Thank you for your thoughtful comment. We fully agree that transparency regarding the scope and inclusion criteria is essential, particularly in the context of the PRISMA 2020 statement. As explained in lines 100–116, we made a deliberate decision to focus solely on GAN architectures that generate samples from noise, rather than covering all possible applications of GANs in cell microscopy. We appreciate the opportunity to elaborate further:

▪ Tasks such as style transfer, image enhancement, or synthetic staining often fall under the category of image-to-image (I2I) translation, which, while impactful, typically require paired datasets to train effectively. In cases where unpaired I2I approaches are used, they often demand significantly larger training sets, which can be difficult to obtain in cell microscopy due to the cost and complexity of data acquisition.

▪ In contrast, generating images from noise does not rely on paired data and is generally more accessible to a broader range of research setups — particularly those with limited data or computational resources. This makes such models especially attractive in microscopy, where data scarcity is a common issue.

▪ Lastly, we chose to limit the scope in order to maintain completeness and analytical depth. Including the full range of GAN applications (e.g., I2I tasks) would have introduced a wide variety of goals and methodological considerations, making it difficult to provide a focused and coherent analysis within a single review.

We have clarified this rationale in the manuscript to better guide readers and to uphold the transparency principle of the PRISMA 2020 guidelines.

---

## [Decision Letter · Decision Letter 4]

Systematic review of generative adversarial networks (GANs) in cell microscopy: trends, practices, and impact on image augmentation

PONE-D-23-26522R4

Dear Dr. Lesmes-Leon,

We’re pleased to inform you that your manuscript has been judged scientifically suitable for publication and will be formally accepted for publication once it meets all outstanding technical requirements.

Kind regards,

Abel C. H. Chen

Academic Editor

PLOS ONE

Additional Editor Comments (optional):

Reviewers' comments:

Reviewer's Responses to Questions

**Comments to the Author**

1. If the authors have adequately addressed your comments raised in a previous round of review and you feel that this manuscript is now acceptable for publication, you may indicate that here to bypass the “Comments to the Author” section, enter your conflict of interest statement in the “Confidential to Editor” section, and submit your "Accept" recommendation.

Reviewer #4: (No Response)

2. Is the manuscript technically sound, and do the data support the conclusions?

Reviewer #4: (No Response)

3. Has the statistical analysis been performed appropriately and rigorously? 

Reviewer #4: (No Response)

4. Have the authors made all data underlying the findings in their manuscript fully available?

Reviewer #4: (No Response)

5. Is the manuscript presented in an intelligible fashion and written in standard English?

Reviewer #4: (No Response)

6. Review Comments to the Author

Reviewer #4: The authors addressed my comments. The manuscript improved after revision.

7. PLOS authors have the option to publish the peer review history of their article (what does this mean?). If published, this will include your full peer review and any attached files.

Reviewer #4: No

---

## [Editor Report · Acceptance letter]

PONE-D-23-26522R4

PLOS ONE

Dear Dr. Lesmes-Leon,

I'm pleased to inform you that your manuscript has been deemed suitable for publication in PLOS ONE. Congratulations! Your manuscript is now being handed over to our production team.

Kind regards,

on behalf of

Dr. Abel C. H. Chen

Academic Editor

PLOS ONE